# Mitochondrial dynamics quantitatively revealed by STED nanoscopy with an enhanced squaraine variant probe

Xusan Yang [1,7,8✉], Zhigang Yang [2,8✉], Zhaoyang Wu [1,8], Ying He[2], Chunyan Shan [3,4], Peiyuan Chai[3], Chenshuo Ma[5], Mi Tian[2], Junlin Teng [3], Dayong Jin [6], Wei Yan[2], Pintu Das[2], Junle Qu [2✉] & Peng Xi [1,6✉]

Mitochondria play a critical role in generating energy to support the entire lifecycle of biological cells, yet it is still unclear how their morphological structures evolve to regulate their functionality. Conventional fluorescence microscopy can only provide ~300 nm resolution, which is insufficient to visualize mitochondrial cristae. Here, we developed an enhanced squaraine variant dye (MitoESq-635) to study the dynamic structures of mitochondrial cristae in live cells with a superresolution technique. The low saturation intensity and high photostability of MitoESq-635 make it ideal for long-term, high-resolution (stimulated emission depletion) STED nanoscopy. We performed time-lapse imaging of the mitochondrial inner membrane over 50 min (3.9 s per frame, with 71.5 s dark recovery) in living HeLa cells with a resolution of 35.2 nm. The forms of the cristae during mitochondrial fusion and fission can be clearly observed. Our study demonstrates the emerging capability of optical STED nanoscopy to investigate intracellular physiological processes with nanoscale resolution for an extended period of time.

[1] Department of Biomedical Engineering, College of Engineering, Peking University, Beijing 100871, China. [2] Key Laboratory of Optoelectronic Devices and Systems of Ministry of Education and Guangdong Province, College of Physics and Optoelectronic Engineering, Shenzhen University, Shenzhen 518060, China. [3] School of Life Sciences, Peking University, Beijing 100871, China. [4] National Center for Protein Sciences, Peking University, Beijing 100871, China. [5] Material Science and Engineering, Rutgers University, Piscataway, NJ 08854, USA. [6] UTS-SUStech Joint Research Centre for Biomedical Materials & Devices, Department of Biomedical Engineering, Southern University of Science and Technology, Shenzhen, China. [7] Present address: School of Applied and Engineering Physics, Cornell University, Ithaca 14853, USA. [8] These authors contributed equally: Xusan Yang, Zhigang Yang, Zhaoyang Wu. ✉email: xy389@cornell.edu; zhgyang@szu.edu.cn; jlqu@szu.edu.cn; xipeng@pku.edu.cn

The term mitochondria comes from the Greek words mito (particle) and chondria (lines), which describe their various structural characteristics in live cells. Their various structural characteristics can reflect a number of cell activities, such as proliferation, migration, and resistance to therapy[1]. Inside mitochondria, ATP is synthesised at the folds in the inner membranes called cristae. The size of a mitochondrion is usually ~1 μm in diameter and 4–10 μm in length, whereas the distance between the cristae in mitochondria is ~100 nm, which means imaging with a resolution far below 100 nm is necessary to visualise the gaps between mitochondrial cristae. However, limited by the diffraction of light, conventional optical microscopy techniques are insufficient to visualise the sub-mitochondrial structures (cristae) and their dynamics[2].

A variety of superresolution techniques have been utilised to visualise submitochondrial structures. For example, Jans et al.[3] discovered the mitochondrial inner membrane organizing system (MINOS) with STED. With 3D-STED, Schmidt et al.[4] imaged mitochondrial cristae with isotropic resolution. Using 3D stochastic optical reconstruction microscopy (STORM), Huang et al.[5] showed the 3D ultrastructure of the mitochondrial network in a fixed cell. Furthermore, to resolve the dynamics of mitochondria in living cells, Shim et al.[6] developed lipophilic cyanine dyes, while Tang et al.[7] developed a photoactivatable Znsalen complex. Ishigaki et al.[8] reported the rich dynamics of mitochondria with a rhodamine derivative. However, because they are limited by spatial resolution, the cristae can only be visualised as lamellar curtain-like structures. In both cases, the spatial resolution was insufficient to distinguish individual cristae for further analysis. Danielli et al.[9] demonstrated label-free third-order photoacoustic (PA) nanoscopy of the cytochromes in the inner mitochondrial membrane in fibroblasts with a resolution of 88 nm. Recently, we reported the use of Hessian structured illumination microscopy (SIM) to investigate mitochondrial dynamics, in which the cristae can be clearly visualised[10]. Even though Hessian-SIM features a fast imaging speed, the highest spatial resolution (~90 nm) is still insufficient to measure the sizes and distances in cristae during their evolution[11]. The high spatial resolution (~50 nm) and temporal resolution (~1 frame per s) of STED make it the most promising choice for the study of mitochondria, which are akin to tiny cells inside the host[3,4,12,13].

Two constraints prohibit STED from being used for long-term live-cell imaging. First, cells have limited tolerance to light exposure due to phototoxicity. Mitochondria are more sensitive to light than other cellular organelles; excessive light exposure can cause mitochondrial dysfunction and mitophagy[14–16]. Second, for STED nanoscopy, high-intensity light is required to achieve improved resolution because the fluorescence in the donut area must be converted to stimulated emission through high-power laser illumination[17]. Moreover, the existing dyes for labelling mitochondria (such as the MitoTracker dyes) have a practical disadvantage, as they are not photostable enough to endure long-term STED imaging at high resolution[8,18]. With the recently developed SNAP substrates and benzylguanine derivative tags, Bottanelli et al.[19] imaged mitochondria and the ER network with two-colour STED for 36 s. The Testa group demonstrated 3D nanoscale imaging of the mitochondrial outer membrane labelled with rsEGFP2-Omp25 (8 vol for 26 s)[20].

To address the challenges in long-term STED live-cell nanoscopic imaging, in this work, we have developed a squaraine dye derivative (MitoESq-635) that is compatible with live cells. The primary advantage of this dye is that it can be easily depleted by a STED laser at relatively low power, which allows the cells to stay in their native state during STED imaging. With this MitoESq-635 STED dye, we achieved a spatial resolution of 35.2 nm during time-lapse imaging of mitochondrial cristae dynamics. The fusion and fission processes can be clearly visualized. Moreover, even the evolution of the cristae structure can be observed over time, which is impossible with other current imaging techniques. Because it benefited from the photostability of MitoESq-635, 3D stack STED imaging of a HeLa cell was demonstrated. Because it enables dynamic imaging of mitochondrial cristae with superresolution, we believe that the MitoESq-635 dye will be widely utilised for the study of mitochondria-related cell behaviours and the discovery of the origin of mitochondria.

## Results

**Imaging live cells with a modified squaraine dye**. We have recently developed an enhanced squaraine variant dye (MitoESq-635) for the labelling of mitochondria in live cells. Its chemical structure is shown in Fig. 1a. It can be excited at 635 nm with 775 nm STED (Fig. 1b). A hexylamidophenylarsenicate moiety is conjugated to the sulfide atom at the central position of the four-membrane ring in the squaraine dye (Supplementary Note 1), which can potentially be used as a protein label for the mitochondrial membrane or other organelles in live cells (Supplementary Note 2). Due to the fast binding of phenylarsenicate to vicinal dithiols and the prioritised targeting of mitochondria by the dye molecules, incubating live cells with MitoESq-635 for a few minutes is sufficient to label the mitochondrial membrane with high density. Vicinal-dithiol-containing proteins (VDPs) containing two active thiol groups in the vicinity can be covalently labelled by a phenylarsenicate conjugate[21,22]. To verify that the enhanced squaraine variant VDP probe (MitoESq-635 in Fig. 1a) is specifically bound to the membrane proteins in the mitochondria, HeLa cells treated with MitoESq-635 for 5 min were imaged with a confocal laser scanning microscope. Imaging to detect MitoESq-635 and MitoTracker colocalization in HeLa cells suggested that the probe was mainly concentrated in mitochondria (Supplementary Figs. 1–3). During colocalization studies with ER and lysosome trackers, some localisation within the ER was also observed[23]. Additionally, MitoESq-635 is widely compatible with different cell lines, such as HeLa, MCF7, RAW 264.7, U2OS and primary neuron cells (Supplementary Fig. 4). The covalent binding of MitoESq-635 with mitochondrial VDPs was further verified by SDS-PAGE, fixed cell washing and colocalization experiments (Supplementary Figs. 5 and 6, Supplementary Note 3).

**Photostability and stimulated emission saturation intensity of the modified squaraine dye**. MitoESq-635 exhibited a maximum molar absorption coefficient in $H_2O$, $0.59 \times 10^5$ L mol$^{-1}$ cm$^{-1}$, at 635 nm. The fluorescence quantum yield (Φx) and lifetime depend on the polarity of the solvents (Supplementary Fig. 7a–c and Supplementary Note 4). MitoESq-635 has an emission peak at 670 nm with a quantum yield (Φx = 0.25) in dimethylsulfoxide (DMSO). Due to the acceleration of the nonradiative decay processes from the lowest excited singlet state, the fluorescence quantum yield and lifetime are quite lower in water and PBS than values of these parameters in organic solvents (e.g., DMSO). The fluorescence lifetime of MitoESq-635 was measured to be 1.7 ns after labelling the mitochondria of HeLa cells (Supplementary Fig. 7d–g). Compared with gold standard live-cell mitochondrial probes, MitoESq-635 exhibits much more robust photostability when exposed to a focused laser during confocal microscopy (MitoTracker Green) and during STED nanoscopy (MitoTracker DeepRed, Fig. 1d, e). With MitoESq-635, rare photobleaching and mitochondrial shape variations are observed upon exposure to STED scanning for over 100 s (1 s per frame, imaging time: 0.66 s, recovery time: 0.34 s, STED beam: 30.2 mW at 775 nm), as shown in Fig. 1e. Under the same imaging conditions, in contrast, the

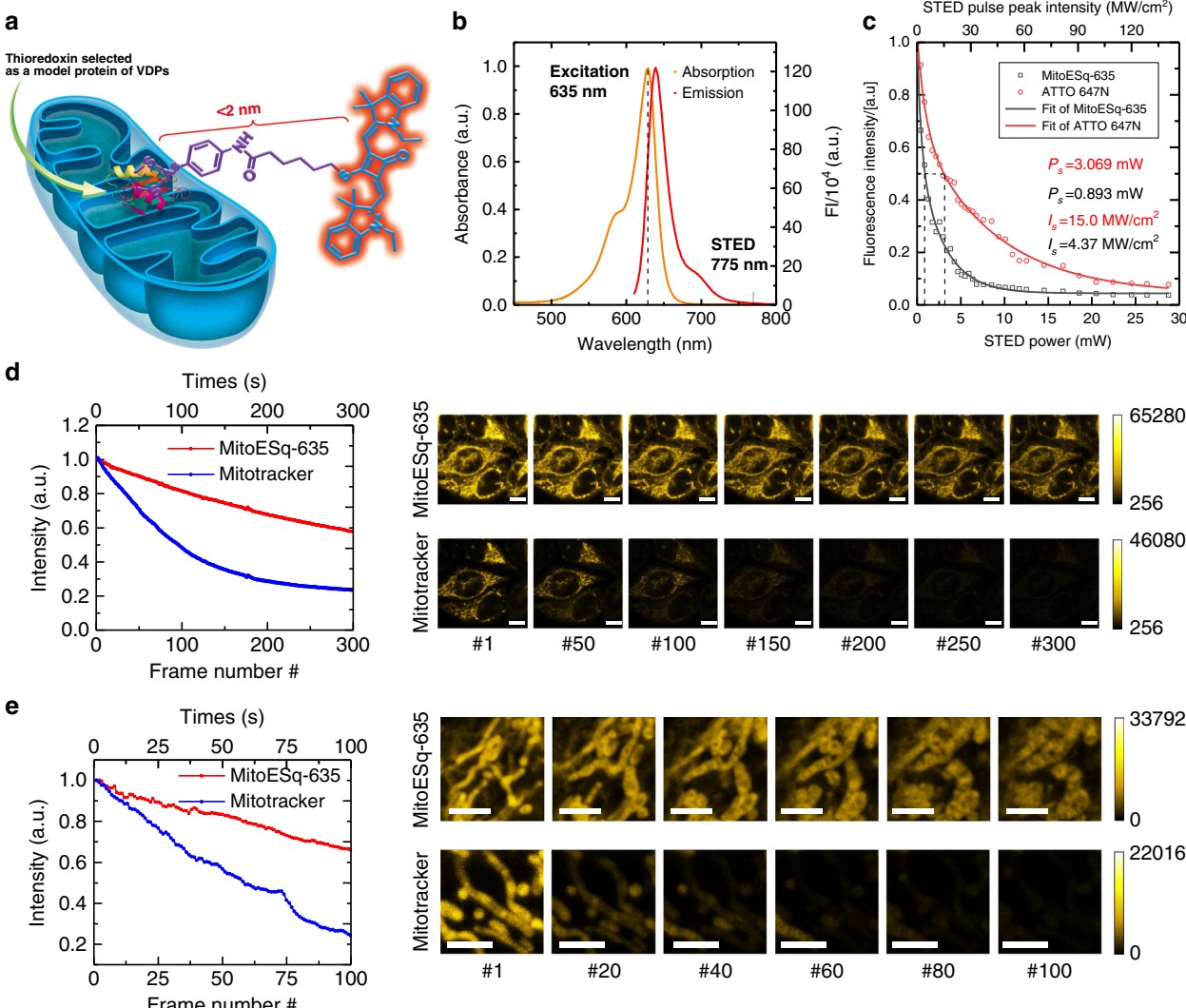

**Fig. 1 A highly photostable, bright, enhanced squaraine probe for mitochondria labelling. a** Chemical structure of MitoESq-635 used for the specific labelling of VDPs. **b** Absorption and emission spectrum of the enhanced squaraine dye, for which a 775-nm pulse laser can be employed for depletion and a 635-nm can be employed laser for excitation during the STED setup. **c** The detected fluorescence signal of the MitoESq-635 and ATTO 647N solution dye pool on a coverslip as a function of the depletion beam intensity; the excitation beam was 635 nm with a 5-ps pulse width and 80 MHz, and the STED beam was 775 nm with 600 ps pulse width and 80 MHz. **d** Comparison of photostability under a confocal laser microscope of HeLa cells co-stained with MitoESq-635 (0.1 μM) and MitoTracker Green (Rhodamine123) (0.1 μM). Upper row, MitoESq-635 excited at 633 nm and collected at 645–680 nm. Lower row, MitoTracker Green (Rhodamine 123) excited at 488 nm and collected at 500–560 nm. Confocal images were obtained under the same imaging conditions, excitation under 1.97 μW averaged power and 1 frame per s acquisition speed (imaging time: 0.66 s, recovery time: 0.34 s). The fluorescence signal of each image is plotted as a function of the recorded image number. Scale bar, 10 μm. **e** Comparison of the photostability under a STED nanoscope of living HeLa cells stained with MitoESq-635 (0.1 μM) and MitoTracker DeepRed (0.1 μM). STED images were obtained under the same imaging conditions: excitation under 1.1 μW averaged power at 640 nm, STED beam of 30.2 mW average power at 775 nm, and 1 frame per s acquisition speed (imaging time: 0.66 s, recovery time: 0.34 s). The fluorescence signal of each image is plotted as a function of the recorded image number. Scale bar, 2.5 μm.

fluorescence signal from MitoTracker Green dropped very quickly due to significant photobleaching (>70% in 100 scans, Fig. 1e), making it unsuitable for long-term live-cell STED imaging. Supplementary Fig. 8 shows a similar result: MitoESq-635 exhibited much more robust photostability than MitoTracker. When the STED power is reduced, data was successfully acquired by using MitoESq-635 with STED imaging over 200 frames for 10 min (3 s per frame, imaging time: 2.58 s, recovery time: 0.42 s, STED beam of 8.96 mW at 775 nm), as shown in Supplementary Movie 1 and Supplementary Note 5.

ATTO 647N is the standard fluorescent label in the red spectral region commonly used for STED nanoscopy because of its strong

absorption, high photostability and high resolution at relatively low STED power. However, the squaraine-STED dye has a saturation intensity of 4.37 MW/cm², which is ~3.4-fold lower than that of ATTO 647N (15.0 MW/cm², Fig. 1c, which is close to the data for 10 MW/cm² provide in the reference study[24]). The MitoESq-635 probe exhibited low toxicity in HeLa cells at a concentration of 1 μM after 1 h of incubation (Supplementary Fig. 9), which suggests that the morphological changes of mitochondria are primarily from exposure to the STED imaging light. However, as the concentration and incubation time increase, probe cytotoxicity will also occur. The low saturation intensity and extended photostability make the dye very suitable for long-term STED imaging in live cells.

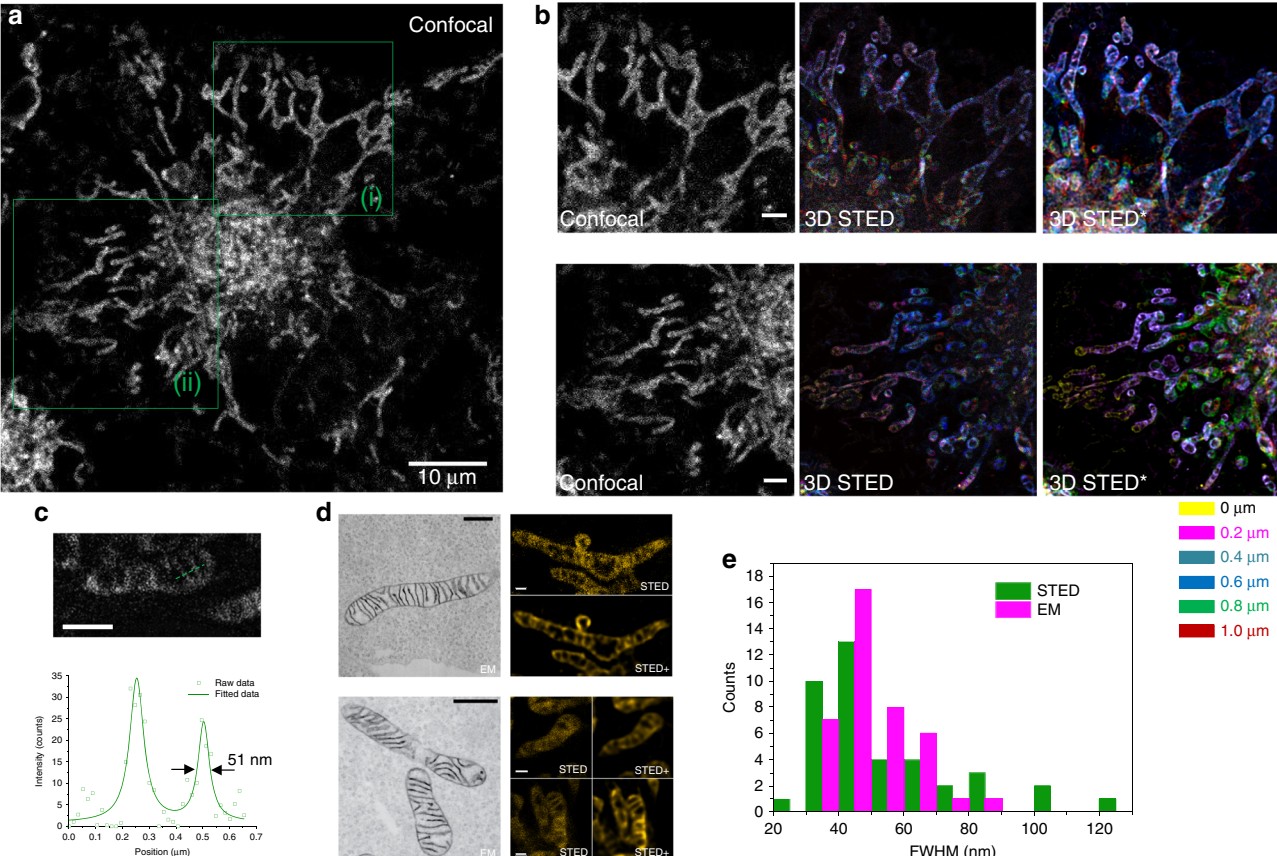

**Fig. 2 3D STED z-stack with superresolution imaging of the mitochondria in HeLa cells with MitoESq-635. a** 2D confocal image of mitochondria labelled with an enhanced squaraine probe in living HeLa cells. **b** Colour-coded STED imaging of the 3D stack (raw data) of the corresponding green boxed area (i, ii) in **a**; the, upper three images are in (i), and the lower three images are in (ii). The image labelled STED asterisk is the corresponding deconvolved image of the STED image; different colours indicate different deaths. **c** A zoom-in view and resolution plot of mitochondrial cristae. **d** In the STED images and the corresponding deconvoluted images (labelled STED plus), the cristae structures in living cells can definitely be resolved and can be compared to the previously reported engineered peroxidase (APEX 2)-labelled electron microscopy (EM) images that were reprinted by permission from Springer Nature Customer Service Centre GmbH: Nature, Nature Methods (Directed evolution of APEX2 for electron microscopy and proximity labelling, Stephanie S Lam et al.), Copyright (2014), (https://www.nature.com/articles/nmeth.3179)[25]; scale bars in EM and STED images are 500 nm and 1 μm, respectively. **e** Cristae width distribution histogram based on STED and EM images. The raw data for the STED images (1024 × 1024 pixels) in different layers of **b** are shown in Supplementary Movies 2 and 3, in which the z-step size is 200 nm, the 635 nm laser used for excitation, and the 775 nm laser used for depletion at the back focal plane of the objective were 5.2 μW and 36 mW, respectively; the pixel dwell time was 4 μs. Scale bars in **a**–**c** are 10, 2 and 1 μm, respectively. The scale bars of the EM and STED fluorescence images in **d** are 500 nm and 1 μm.

Next, the probe was employed for 3D z-stack STED imaging in live cells. Previously, it was challenging to perform STED in 3D z-stacks because of photobleaching during imaging. As shown in Fig. 2 (Supplementary Movies 2 and 3), the regions (i, ii) in panel a were scanned with a z-step of 200 nm, and STED 3D z-stacks were obtained by the construction of different layers of STED images represented by different colours. Raw data were used to obtain fine STED images of mitochondria, which could be optimised through deconvolution. From the magnified STED image of a single mitochondrion, it can be observed that the probe was mainly localised in the cristae, which were formed by inner membrane folding within the mitochondria. The morphological structures in cristae were resolved by STED nanoscopy and matched well with those in previously reported APEX 2-labelled electron microscopy (EM) images (panel d)[25], and their full width at half maximum was as small as ~51 nm (panel c), which was in line with that observed in the EM images (panel e). To avoid measurement artefacts caused by low signal-to-noise ratios, we also performed Fourier ring correlation (FRC) analysis[26]. The FWHM here is close to the FRC analysis result (as shown in Supplementary Fig. 10).

**Subcellular dynamic nanoscopic imaging with MitoESq-635.** While detailed structural information for intracellular membranes at the nanoscale has emerged from electron microcopy[27–29] as well as localised superresolution microscopy[30], it is still unclear how different forms of mitochondria have evolved. A superresolution technique with sufficient spatial and temporal resolution is highly desired. To study the highly dynamic and subtle morphological changes within mitochondria, we developed a long-term STED nanoscopic imaging strategy to visualise mitochondrial membranous dynamical structures in living cells. Figure 3a (Supplementary Movie 4) shows the mitochondrial dynamics in a living Drp 1 KO HeLa cell visualized with STED (imaging time: 2.58 s, dark recovery: 0.42 s, 3 s per frame in total; STED beam: 7.84 mW at 775 nm before the objective). Figure 3b, c show the change in the FWHM value in the STED image at different time points from 71 nm at 00:00 to 132 nm at 03:00. The time-lapse STED images of living cells will have both fluorescence signals and resolution loss as the frame number increases. Note that due to its low SNR, the resolution measurement in frame 60 has difficulty exhibiting a reliable cross-section profile and determining the resolution relying on its FWHM, so we also

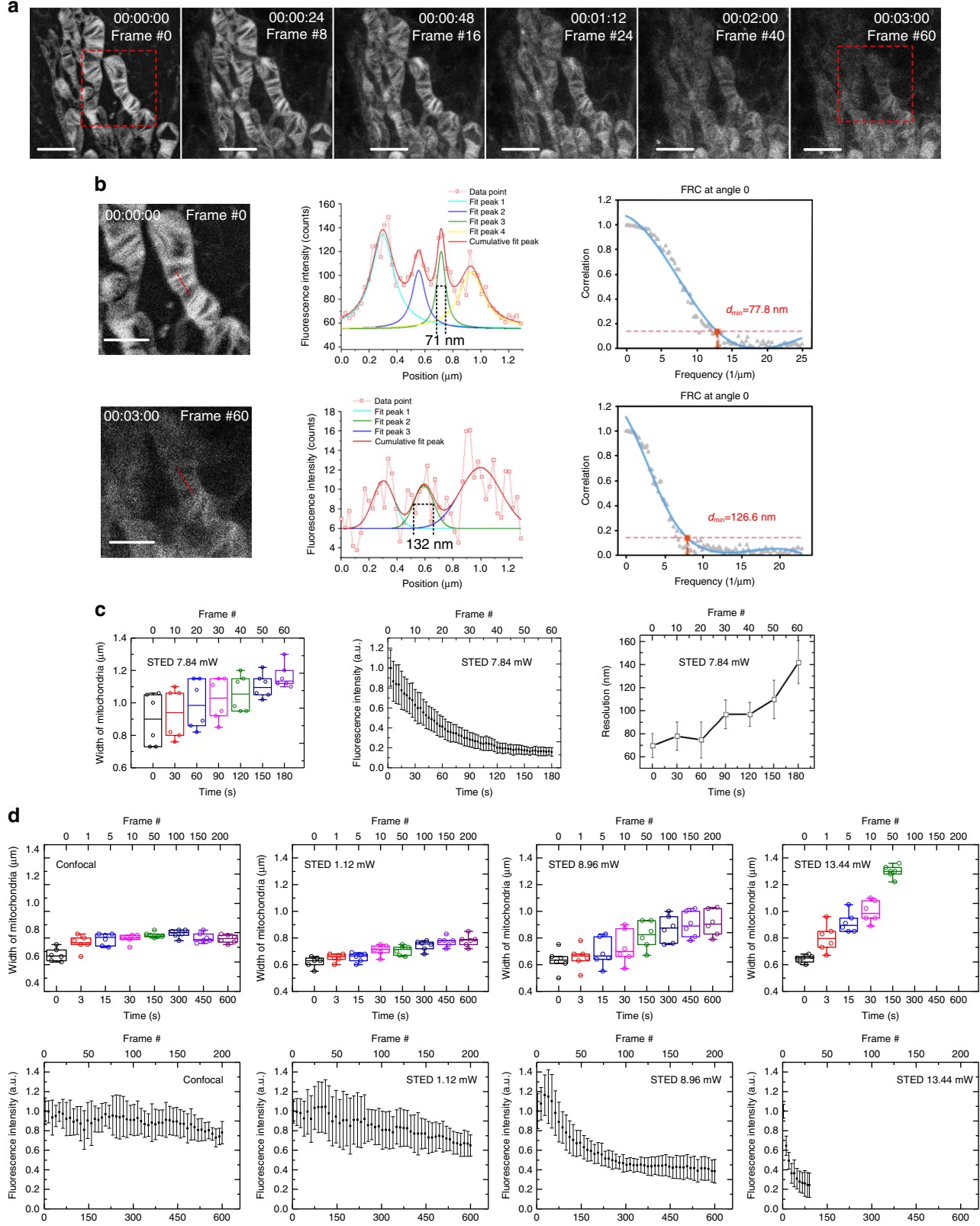

**Fig. 3 STED imaging of mitochondrial cristae and the subsequent statistical analysis. a** Time-lapse STED imaging of the mitochondria. Scale bar, 3 μm. **b** The enlargement of the red box in **a**, cross-section intensity profile at the red line and their FRC analysis results regarding the resolution[26]. Scale bar, 2 μm. **c** Time-lapse plots of mitochondrial width, fluorescence intensity and image resolution. **d** Changes in mitochondrial width and fluorescence intensity under different STED intensities. Where box plots are shown, the centreline represents the median, the bounds of the box represent the upper and lower quartiles, the whiskers represent minimum and maximum values, and the hollow circle shows dot plots, $n = 6$. Error bars represent the standard deviation of the means, $n = 6$.

included the FRC analysis results of frames #1 and #60. Figure 3c shows that mitochondria width increases and that intensity decreases with time. In Fig. 3d, the changes in mitochondria width and fluorescence intensity under different STED powers are further analysed. The width and fluorescence intensity change faster when the STED power is higher. The 775-nm STED power of 7–9 mW should be an optimised power in our case, balancing between resolution, photobleaching and photodamage (Supplementary Note 6). To further improve resolution, STED power should be increased, in which case a longer dark recovery can help reduce photobleaching and photodamage. Taking advantage of the live-cell compatibility of MitoESq-635, the rapid nanoscale spatiotemporal mitochondrial dynamics in living HeLa cells were captured with STED (3 s per frame; exposure time 2.58 s; rest for 0.42 s; STED power at 8.96 mW, 7.84 mW and 6.72 mW; 200 frames; 10 min; Supplementary Movie 1). Additionally, we were able to record the longer-term (over 60 min, 3.9 s per frame followed by 71.5 s of dark recovery) nanoscale spatiotemporal dynamics of the mitochondrial inner membrane with STED nanoscopy at a resolution of 35.2 nm (Fig. 4a–c, for the raw data, Supplementary Movie 5; this resolution is close to the FRC result shown in Supplementary Fig. 11) and observe mitochondrial fission. As shown with the white arrow, the thin and elongated mitochondria first form bubble structures at 16:22; then, the cristae grow quickly inside these bubbles, and they separate to form individual small mitochondria with rich and constantly changing internal cristae structures (21:22–56:33) (Fig. 4a).

The time-lapse STED images revealed thin, extended tubular intermediates connecting neighbouring mitochondria both before and after fission (Fig. 4d). The average widths of these tubular structures before and after fission are ~85 and 42 nm, respectively. Additionally, the cristae width after fission was measured to be 42 nm, which agrees well with the reported EM results[31]. The quenching beam (STED) could cause intracellular condition changes at a light dose to a certain extent. Upon further increasing the power of the quenching beam, the fluorescence molecule could be oxidised under the high depletion beam power of 51 mW (Supplementary Fig. 12).

We also observed the fusion process of mitochondria, as shown in Fig. 4f, h (Supplementary Movies 6 and 7). Figure 4g shows the width of mitochondria at the fusion point in Fig. 4f, which grows from ~0.3 μm to ~0.8 μm. As highlighted by the white dashed boxed area in Fig. 4h, the mitochondria were initially in a typical line structure. Then, the cristae began to form and grow rapidly (15:05) as well as merge and combine with a larger area (23:53-41:28). The merge made the finger-like structures disappear, and a hollow structure with several shoots containing cristae formed (52:47–60:19). The shoots then formed a complicated network of cristae (44:00). The arrows point to several areas where the rod-like isolated mitochondria expanded with newly formed cristae and fused into one large mitochondrion with several bubbles. After mitochondrial fusion, the cristae membrane began to remodel. All of these changes are supported by previous EM results (Supplementary Fig. 13). The inner mitochondrial membrane is compartmentalised into numerous cristae, which expand the surface area of the inner mitochondrial membrane to enhance its ability to produce ATP. The oxidative phosphorylation system (OXPHOS) facilitates energy conversion for ATP production so that mitochondria can supply energy for other subcellular organelles, which make mitochondria the indispensable 'power plants' for eukaryotic cells.

## Discussion

Mitochondria are vital subcellular organelles because they generate energy for a variety of cellular and developmental processes. Mitochondrial dysfunction is associated with numerous severe diseases, including several devastating neurodegenerative diseases. Furthermore, the degree, functional relevance and molecular causes of the heterogeneity of mitochondrial structure and function in healthy and stressed single cells require both molecular/biochemical tools and advanced superresolution living cell microscopy. When responding to different cellular statuses, mitochondria constantly change the morphologies of their outer and inner membranes to regulate energy production. To investigate their dynamic behaviour in live cells, fast and high-resolution superresolution microscopy techniques are required. Because the distances between mitochondrial cristae are <90 nm, SIM cannot fully resolve cristae due to limited resolution enhancement. Single-molecule localisation microscopy techniques can provide 20–50 nm spatial resolution, but their poor temporal resolution can lead to significant motion blur during the imaging process. Although STED can provide high spatial and temporal resolution simultaneously, the current STED dyes for mitochondria are generally incompatible with live cells during long-term STED imaging due to high light intensity-induced photobleaching. Very recently, during the review process for this manuscript, the Jakobs group reported a cell line expressing mitochondrial protein fused to a SNAP-tag to enable high resolution STED of mitochondrial cristae in live cells for 2 min (every 15 s per frame, 8 frames)[32]. Wang et al.[33] developed a fluorescent labelling reagent and captured the structures of the mitochondrial cristae with a resolution of ~60 nm when depleted at 660 nm for 390 s. In the two works, the Jakobs group achieved high resolution (70 nm) STED of mitochondrial cristae in live cells for 2 min, and Wang et al. captured the structure of mitochondrial cristae with a resolution of ~60 nm (after deconvolution) for 390 s (300 frames). Compared with Ref. [32], our technique offers a much longer imaging time. With respect to Ref. [33], we obtained better resolution with less STED power. A detailed comparison can be found in Supplementary Tables 3 and 4 (Supplementary Note 5). In this work, we report the use of MitoESq-635, which is a squaraine dye specifically designed for long-term live-cell STED imaging of mitochondria. MitoESq-635 probes have superior photophysical properties compared with organic dye ATTO 647N probes, which are commonly employed in STED nanoscopic imaging. The colocalization of MitoESq-635 with MitoTracker demonstrated that it can label mitochondria membranes. Here, we achieved 35.2 nm spatial superresolution. The dynamic imaging of live cells over 50 min (3.9 s per frame followed by 71.5 s of dark recovery) clearly revealed the fusion and fission processes of mitochondria. Because of the low saturation intensity, STED imaging of 3D stacks can reveal the ultrastructure of mitochondria in live cells. Overall, the labelling specificity and performance of MitoESq-635 in low-power STED make it highly attractive as a next-generation standard for long-term superresolution imaging of mitochondria.

## Methods

**UV/Vis absorption and fluorescence spectroscopy measurements**. A stock solution of MitoESq-635 (1 mM) was prepared in DMSO solvent and then diluted with ethanol to 1.0 μM. The UV/Vis absorption spectra of MitoESq-635 were measured using a Perkin Elmer 3000 spectrophotometer (PE, USA), and fluorescence emission was measured with Horiba spectrofluorometric equipment (Horiba) with a Xenon lamp, a regular PMT detector and an In/Ga/As detector for NIR II measurements.

**Fluorescent probe design**. Synthesis procedures of compounds and design strategies are shown in Supplementary Note 1 (Supplementary Fig. 14) and Supplementary Note 2 (Supplementary Fig. 15), respectively. The probe is suitable for live-living cell imaging (Supplementary Fig. 16). Supplementary Figs. 17–23 show chemical structure identifications ($^1$H, $^{13}$C NMR and mass spectra) of compounds and MitoESq-635.

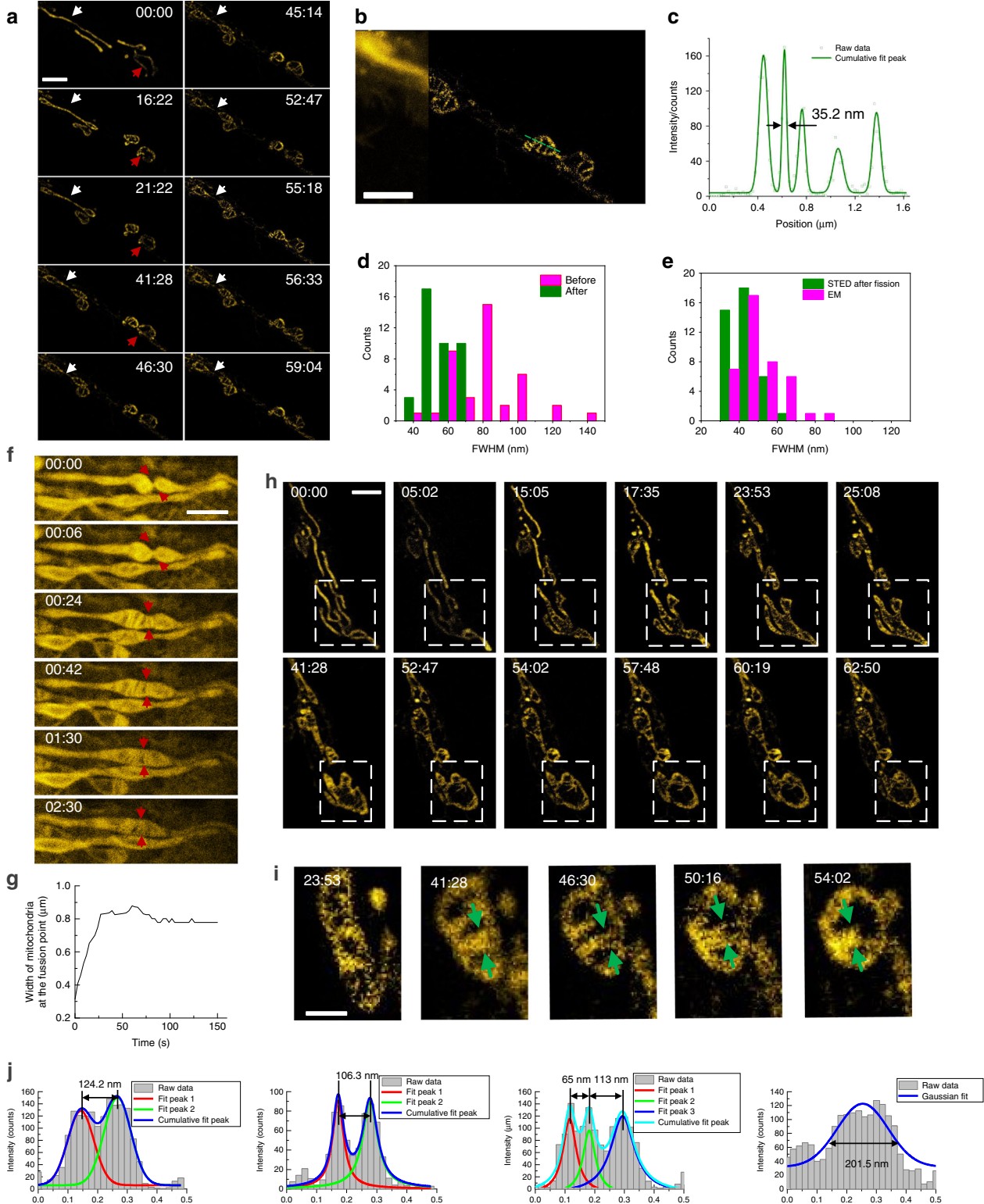

**Fig. 4 Mitochondrial fission and fusion process observed with STED optical nanoscopy. a** Time-lapse STED imaging of mitochondria. The white and red arrows show typical mitochondrial fission processes. **b** A comparison of confocal (left) and STED (right) imaging. The intensity profile across the green dashed line in **b** is shown in **c**. A 35.2-nm resolution can be obtained from STED superresolution microscopy. **d** Distribution of cristae widths. Green bars: connective tubule width distribution histogram after fission. Pink bars: tubule width distribution histogram before fission. **e** Cristae width after fission distribution histogram in STED and EM. **f** The red arrows show typical mitochondrial fusion processes, 3 s/frame. **g** The width of mitochondria changed with time at the fusion point in **f**, as indicated by the red arrows. It is 3 s per point, 51 points in total and illustrated by Origin software. **h** Quantitative images of mitochondria revealed cristae dynamics during fusion. **i** Cristae dynamics from a line shape to a bubble shape. **j** The cross-section plot along the green arrows in **i** is shown. Scale bars in **a**, **b**, **f**, **h** and **i** are 2, 2, 2, 5 and 1 μm, respectively.

**Cell culture**. The cell lines, including HeLa, MCF7, RAW 264.7, U2OS, SH-SY5Y and SKOV cells, were cultured at a suitable density (moderate) in DMEM media. All culture media were supplemented with penicillin (100 units/mL), 10% (v/v) FBS (WelGene) and streptomycin (100 mg/mL), and cells were incubated in a 5% $CO_2$ atmosphere at 95% humidity and 37 °C. The cells were placed in confocal dishes 24 hours prior to the experiments.

**Confocal microscopy**. The selected cell lines were seeded on cover slips or glass-bottomed dishes (SPL Lifesciences Co., Ltd.) before they were grown to a suitable density (24 hours) via incubation in a humid atmosphere containing 5% (v/v) $CO_2$ at 37 °C. The cells were incubated with MitoESq-635 or mitochondrial, lysosomal, or ER trackers for different times (e.g., 30 min) in a humid atmosphere with 5% (v/v) $CO_2$ at 37 °C. Then, the cell images were obtained using a confocal laser scanning microscope (Leica SP2 and SP8, Leica, Germany). Related information is available in the figure captions.

**SDS-PAGE and fluorescent imaging of gels**. The selectivity of MitoESq-635 for proteins and cells was identified by 10% SDS-PAGE experiments. The different protein samples were treated with DTT or $H_2O_2$ and incubated with MitoESq-635 in PBS buffer at 37 °C for 1 h. After labelling, the obtained samples were precipitated in 50% (v/v) acetone for 2 h at −20 °C and then mixed with SDS-PAGE loading buffer containing tris (2-carboxyethyl) phosphine (TCEP), and electrophoresis was carried out immediately. The gel was observed and imaged using a fluorescent scanner (Tanon-5220S, Shanghai, China) with green light excitation with a bandpass filter with a range from 635 to 675 nm.

**STED superresolution for live cells**. STED imaging was performed with a Leica TCS SP8 STED 3X system equipped with a white light laser for excitation and a 775-nm pulsed laser for STED depletion. A ×100 oil-immersion objective (Leica, N.A. 1.4) was employed.

**Statistical analysis**. The width of mitochondria and fluorescence intensity were detected by using Fiji software. Both a box plot chart of the mitochondria width and an error bar chart of fluorescence intensity were obtained using OriginPro 9.1 software. The Gaussian cumulative fit peak was processed with Origin 2018 software.

**Deconvolution**. Deconvolution was performed by using Huygens Software embedded in the Leica TCS SP8 STED 3X system. The deconvolution process was completed by using an auto setting in Huygens. Detailed parameters were as follows: (1) background: automatic estimation, (2) estimate mode: lowest, (3) area radius: 0.7, (4) deconvolution algorithm: CMLE, (5) maximum iteration: 40, (6) signal-to-noise ratio: 7, (7) quality threshold: 0.05, (8) iteration mode: optimised, (9) bleaching correction: if possible, (10) PSFs per brick: one PSF and (11) brick layout: auto.

**Statistics and reproducibility**. Figures 1d, e; 2a–d; 3a, b; and SI Figs. 4, 5, 7d, e, 8a, b, 10a, 11a and 12 were repeated two times with similar results. Figure 4a, b, f, h, i and SI Figs. 1–3 and 13 were repeated three times with similar results.

**Reporting summary**. Further information on research design is available in the Nature Research Reporting Summary linked to this article.

## Data availability
The data that support the findings of this study are available through "figshare.com" with the identifier(s) "https://doi.org/10.6084/m9.figshare.12252842"[34]. The data that support the findings of this study are available from the corresponding author upon reasonable request. The MitoESq-635 dye can be obtained from the author (Z.Y.) upon reasonable request.

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

## Acknowledgements

This work was supported by the National Key Research and Development Program of China (2017YFC0110202), the National Natural Science Foundation of China (61525503, 61729501, 61875131, 61620106016, 61835009 and 31971376), the Beijing Natural Science Foundation (JQ18019), the (Key) Project of Department of Education of Guangdong Province (2015KGJHZ002 and 2016KCXTD007), the Guangdong Natural Science Foundation Innovation Team (2014A030312008) and the Shenzhen Basic Research Project (JCYJ20170818100931714, JCYJ20180305125549234 and JCYJ20170412105003520). We thank the National Center for Protein Sciences at Peking University in Beijing, China, for the assistance of providing the Leica SP8 STED commercial system. We would like to thank Professor Antony Chen, Professor Luke Lavis, and Dr. Xu Zhang for their helpful discussions and suggestions.

## Author contributions

X.Y., Z.Y., J.Q. and P.X. initiated the project. X.Y., Z.Y. and P.X. designed the dye and the experiments. X.Y., Z.W., C.S., P.C. and Y.H. prepared living cells labelled with the enhanced squaraine probe and performed the STED superresolution experiments in living cells. X.Y., Z.W., C.M. and P.X. constructed the microscope setup and designed the optical system for the dye property measurements. M.T., J.T., D.J., W.Y. and P.D. participated in effective discussions. X.Y., Z.Y., Z.W. and P.X. analysed the data. J.Q. and P.X. supervised the project. X.Y., Z.Y. and Z. W. prepared all the figures, X.Y., Z.Y., Z.W. and P.X. wrote the manuscript with contributions from all authors.

## Competing interests

The authors declare no competing interests.
