## [Peer Review File · Nature Communications]

Reviewers' comments:

Reviewer #1 (Remarks to the Author):

With recent advances in super-resolution microscopy, and the increasingly recognized importance of mitochondrial dynamics in mediating mitochondrial function, the manuscript by Yang et al., is timely. The work describes a novel fluorescent probe that should facilitate super resolution microscopy of mitochondria via STED. In particular, the ability to watch cristae dynamics during fission and fusion events is very exciting. While the STED images of mitochondrial cristae are certainly beautiful, and it appears that the new dye described here may be superior to those used in previous published work looking at mitochondrial cristae using STED, I do have some concerns with the manuscript that would need to be addressed.

Specific Issues:

The section describing how the Mito-ES1-635 dye labels mitochondria is confusing. The authors state that it 'labels the plasma membrane with high density'. Then the authors say that the signal is 'concentrated in the eccentric perinuclear regions', but I'm not sure what they mean by these statements and how they are relevant to mitochondrial staining. Clearly the dye does co-localize to mitochondria based on the images shown, but the description in the text is not clear.

The authors claim that the Mito-ES1-635 dye is covalently binding with mitochondrial VDPs. However, in Sup Fig 3, they show that MitoESq-635 binds to TRX-1, a cytosolic protein. Based on the evidence provided, it is unclear to me that the dye is in fact binding specifically mitochondrial membrane proteins, which I think is what the authors are trying to say.

I'm not sure that I agree with the authors that there is no phototoxicity. They describe changes in the mitochondria, 'bubble structures', with cristae growing inside, which I would interpret as swelling of the mitochondria. In particular, the changes in fig 3 between the first frame and the final frame look like swelling and fragmentation of the mitochondria, which is you would expect to see with phototoxicity. If this was the 'normal' state of mitochondria, it should be evident from the start of the imaging, not just later on.

This swelling also appears evident in Fig 4, as the width of the mitochondria appears to increase significantly over time, and not just due to fusion. While it appears that they can in fact catch fusion events, due to the swelling, I'm not sure how well the cristae dynamics they observe reflect what normally happens during fusion.

In order to accurately compare the Mito-ES1-635 dye to other dyes used to image mitochondrial cristae, they should show their own data using their scope and settings, rather than pulling numbers from other publications, as settings and laser power are often unique to each microscope.

General comments:

I would completely remove the first paragraph, which I feel is unnecessary. The evolutionary history of mitochondria and their role in cancer are not really relevant to the current work.

Though mutations in OPA1 cause an optic atrophy phenotype, OPA1 does not encode a photoreceptor, and this has nothing to do with sensitivity to light or phototoxicity.

The authors need to include a bit more information regarding the deconvolution analysis they are applying to their images.

Regarding the average widths of cristae they use 42nm and 43nm for the width following fission. I assume this should be the same number.

The authors state 'the fluorescent molecule could melt', with high laser power. Not sure that 'melt' is the correct term here.

While the images in Sup Fig 2 certainly look like mitochondrial networks, it would be nice to see co-localization with another mitochondrial marker for these other cell lines.

Generally speaking, many of the images are too small to adequately see at 100% magnification.

The co-localization images in Sup fig 1 are not great.

For the supplementary videos 5 and 6, it would be nice to see the white dashed boxed areas shown in Fig 4 in order to focus the attention to the correct position.

Reviewer #2 (Remarks to the Author):

The manuscript by Yang and coworkers addresses a timely need in super-resolution imaging, namely the generation of live-cell imaging compatible probes to follow mitochondrial dynamics. The topic per se is thus relevant and interesting to a wide readership. A very similar paper appeared in Proceedings of the National Academy of Sciences earlier this year (Wang et al., PNAS 116, p15817 (2019)). In that paper, also a mitochondria-selective dye was presented for STED imaging, allowing observation of cristae dynamics. Therefore, claims of novelty such as "for the first time" in the abstract and elsewhere are problematic and that prior work should be acknowledged. Imaging in the far red as in the present manuscript (STED laser at 775 nm) is desirable to reduce interactions with cellular constituents. In terms of imaging performance, I would rate the PNAS paper higher (albeit at 660 nm STED wavelength) and I would also rate a recent paper by the Jakobs group higher in terms of imaging quality, using SNAP tag and silicon rhodamine for depletion at 775 nm (Stephan et al., Sci Rep 640920 (2019)). Therefore, the specific merit of the manuscript is to provide a label for STED microscopy at 775 nm which has affinity for mitochondria.

However, the manuscript has several shortcomings, such that I cannot recommend it for publication in its present state. Several of the claims are not sufficiently supported by the quality of the data. In other instances the information provided is insufficient to truly assess the validity of the measurements.

Please find my detailed assessment below.

Major points:

- 1) Absorption coefficient, quantum yield, fluorescence lifetime etc. for the squaraine dye should be given.
- 2) Reporting of imaging parameters and timing of recordings is insufficient, making it either impossible to gauge the true merits or rendering statements misleading. E.g. the section on STED imaging in the Methods section states that each frame was obtained at 1 s and the image series were obtained at a 1-minute interval but it is not clear to which measurements this refers. Precise timing information is warranted in each image caption and detailed imaging parameters should be stated for each measurement. Similarly, all other experiments should be described in sufficient detail to reproduce them.
- 3) In the abstract the authors state that live imaging over 50 minutes is possible. This is a meaningless statement if the actual time it takes for one frame and the intermittent (resting) times are not stated. As far as I understand, this statement refers to Suppl. Video 4, where each frame took about 4 seconds and then a recovery period of about 70 seconds was used, yielding 40 images in total. The authors should rather state the number of images that were taken.
- 4) The authors claim a resolution of 35.2 nm. This statement rests on a single line profile shown and a histogram without any detail on how data was analysed. From the supplementary videos and

the image material provided, I am very doubtful that signal-to-noise ratio is sufficient to claim 35 nm resolution or even 35.2 nm. The authors are encouraged to refer to the relationship of resolution and signal-to-noise ratio in STED microscopy, discussed e.g. by the Vicidomini group (Tortarolo et al., Evaluating image resolution in stimulated emission depletion microscopy. *Optica*, 5, p32 (2018)). Also the resolution determination in Fig. 2c is problematic due to insufficient signal-to-noise ratio. From most of the images provided, I find it hard to clearly delineate individual cristae. In comparison, cristae stand out very clearly in the Scientific Reports paper from the Jakobs group.

- 5) I am somewhat concerned with the specificity of labeling for mitochondria. In Suppl. Fig. 1, there seems to be strong correlation with ER, which is, however, mentioned in the main text.
- 6) p5: The description of Suppl. Fig. 6 in the main text implies that no toxicity arises during 1 h of STED imaging. However, in Suppl. Fig. 6 it is not clear whether STED imaging has been applied in addition to the incubation. It is not described what the viability assay is.
- 7) p 5: The statement on saturation intensity is contradictory. This measurement is not described properly. From the data provided it is not clear that this measurement was done in a way that excludes e.g. the possibility that bleaching might influence the measurement outcome. While superior imaging performance over Atto647N would be an achievement, this statement would need to be supported by further data.
- 8) p5: I find the wording on 3D imaging somewhat misleading. The authors apply only resolution enhancement in the focal plane, not along the optical axis. Taking a stack of 5 STED images with xy resolution enhancement and confocal resolution axially is not particularly challenging. Therefore, also the statement in the conclusion "STED imaging of 3D stacks can reveal the ultrastructure of mitochondria in live cells" is not justified for the measurements presented in the manuscript.
- 9) p7: The authors observe formation of bubbles and fission of mitochondria in long-term STED imaging. It should be checked that these dynamics are not related to the photo-burden inflicted by the imaging.
- 10) Suppl. Note 3: No data is shown to back up the claims that the squaraine dye has wide applicability.
- 11) The Supplements, in particular Suppl. note 4 seem to be an unedited draft version with a series of problematic statements. For example, the statement on acquisition speed is not tenable in this unspecified way in view of the work by Schneider et al., *Nature Methods* 12, p827 (2015) where 125 frames per second super-resolution imaging rate was achieved.
- 12) Suppl. Fig. 5: If MitoESq-635 is covalently linked to inner membrane proteins, it is no surprise that it stays bound also with prolonged imaging. This does, however, not imply that the mitochondrial membrane potential is till intact.

Minor points:

- 1) p2: Wording is unfortunate: "ATP, which is the only energy form that cells can absorb". ATP is certainly the most prevalent energy carrier, but it's not absorbed by the cells but rather synthesized from ADP. Also there are other high-energy bonds in cells that can be used to drive biochemical processes.
- 2) p 2: I am not sure that mitochondrial dynamics "regulate" cancer cell activity.
- 3) p2: The statement on the sampling theorem is unfortunate. Resolution is defined as the capability to discern two nearby structures.
- 4) p2: There is an inconsistency with the author name in the citation Dannie et al..
- 5) Several of the citations are incomplete.
- 6) p3: It would be helpful to include a statement on the absorption wavelength range of OPA1.
- 7) p3: "Stimulated emission" is not a laser process. "Laser" means "light amplification by stimulated emission of radiation". Stimulated emission is an important component of the laser process, but not all stimulated emission is equivalent to lasing.
- 8) p3: "...the efficiency is generally too low to generate effective STED processing." This sentence needs rewording.
- 9) p4: I am not sure whether the authors want to state that they label the "plasma membrane".
- 10) Fig 1c: The authors plot power, not "intensity", as stated in the figure caption.

- 11) In Suppl. Fig. 1, the last column is not adequately described.
- 12) p 8: What do the authors precisely mean by "melting" of dye molecules?
- 13) Fig. 4: What is a "skeletal image"?
- 14) The data in Suppl. Fig. 4 is consistent with covalent labeling, but not a "proof".
- 15) Suppl. note 1, line 66: "literature procedure" would warrant a citation.

Reviewer #3 (Remarks to the Author):

The paper titled "Mitochondrial dynamics quantitatively revealed by STED nanoscopy with an enhanced squaraine variant probe" reported a new fluorescence living cell probe, MitoESq 635. It is an excellent candidate for live cell STED nanoscopy, which will make real impacts on pushing super resolution microscopy towards practical live cell STED imaging. The manuscript is well written and timely. With much better performance over the existing mitochondrial probes, the authors obtained beautiful super resolution images of mitochondrial ultrastructure in 3D and long term (about 1 hour) using STED nanoscopy. They got mitochondrial cristae dynamical videos with a decent resolution of sub-50 nm for 1 hour for the first time in super resolution microscopy and mitochondrial research field. I have several minor questions for the authors to address before it is published on Nature Communications.

1. In figure 1b, I suggest the authors to add the labels of excitation and STED wavelengths.
2. In figure 2c, figure 3c and figure 4d, the details of how the data points were fitted should be described.
3. In Supplementary Figure 9, all the scale bars on fluorescence images were missing.
4. Although this probe worked pretty well with STED nanoscopy, the performance of this probe in live cell STED could be further improved if it can be combined with recent new STED technologies such as RESCUE STED, MINFIELD, DyMIN, STEDD, et al. Can the authors discuss about this?

Response Letter

Manuscript ID: NCOMMS-19-16887-T

Title: Mitochondrial dynamics quantitatively revealed by STED nanoscopy with an enhanced squaraine variant probe

Reviewers' comments:

Reviewer #1 (Remarks to the Author):

With recent advances in super-resolution microscopy, and the increasingly recognized importance of mitochondrial dynamics in mediating mitochondrial function, the manuscript by Yang et al., is timely. The work describes a novel fluorescent probe that should facilitate super resolution microscopy of mitochondria via STED. In particular, the ability to watch cristae dynamics during fission and fusion events is very exciting. While the STED images of mitochondrial cristae are certainly beautiful, and it appears that the new dye described here may be superior to those used in previous published work looking at mitochondrial cristae using STED, I do have some concerns with the manuscript that would need to be addressed.

Specific Issues:

- (1) The section describing how the Mito-ES1-635 dye labels mitochondria is confusing. The authors state that it 'labels the plasma membrane with high density'. Then the authors say that the signal is 'concentrated in the eccentric perinuclear regions', but I'm not sure what they mean by these statements and how they are relevant to mitochondrial staining. Clearly the dye does co-localize to mitochondria based on the images shown, but the description in the text is not clear.

Response:

Thanks for the reviewer's recognition on the importance of this research and our results. We are sorry for the unclear descriptions in the main text. We have removed the sentence "The fluorescence signals are concentrated in eccentric perinuclear regions, whereas little signal is presented in cytoplasm and nucleus regions."

As shown in the new Supplementary Figure 1, 2, 3 the dye does co-localized to mitochondria. Here we also compared Mito Esq-635 with Mito Tracker Green below in several different cell lines, including U2OS (Supplementary Figure 1), HeLa (Supplementary Figure 2, SH-SY5Y and SKOV cells (Supplementary Figure 3). Hence, we can confirm that the Mito Esq-635 probe are targeted efficiently to mitochondria.

Supplementary Figure 1 Co-localization experiment in U2OS cells employing Mito Tracker Green as golden standard for mitochondrial marker. 1st column: enhanced squaraine dye (20 nM) labeled U2OS cells. 2nd column: Mito Tracker Green (50 nM) labeled U2OS cells. 3rd column: merged images. 4th column: 2D intensity histogram and Pearson's coefficients. Excitation wavelength: Mito ESq-635 (640 nm), Mito Tracker Green (490 nm); detect range: MitoESq-635 (660-740 nm), Mito Tracker Green (495-575 nm). Scale bar, 5 μ m.

Supplementary Figure 2 Co-localization experiment in HeLa cells employing Mito Tracker Green as golden standard for mitochondrial marker. 1st column: enhanced squaraine dye (20 nM) labeled HeLa cells. 2nd column: Mito Tracker Green (50 nM) labeled HeLa cells. 3rd column: merged images. 4th column: 2D intensity histogram and Pearson's coefficients. Excitation wavelength: Mito ESq-635 (640 nm), Mito Tracker

Green (490 nm); detect range: MitoESq-635 (660-740 nm), Mito Tracker Green (495-575 nm). Scale bar, 5 μ m.

Supplementary Figure 3 Co-localization experiment in SH-SY5Y and SKOV cells employing Mito Tracker Green as golden standard for mitochondrial marker. 1st column: enhanced squaraine dye (20nM) labeled SH-SY5Y and SKOV cells. 2nd column: Mito Tracker Green (50nM) labeled SH-SY5Y and SKOV cells. 3rd column: merged images 4th column: 2D intensity histogram and Pearson's coefficients; excitation wavelength: Mito ESq-635 (640 nm), Mito Tracker Green (488 nm); detect range: MitoESq-635 (700/75nm nm), Mito Tracker Green (525/50 nm). Scale bar, 10 μ m.

(2) The authors claim that the Mito-ESq-635 dye is covalently binding with mitochondrial VDPs. However, in Sup Fig. 3, they show that MitoESq-635 binds to TRX-1, a cytosolic protein. Based on the evidence provided, it is unclear to me that the dye is in fact binding specifically mitochondrial membrane proteins, which I think is what the authors are trying to say.

Response:

Thanks for your comments here. In Supplementary Fig. 3, we performed SDS-PAGE experiment to prove the probe Mito-ESq-635 can covalently bind to the proteins with two vicinal thiol residues (VDPs), as there is a phenylarsenate moiety in the probe. Thioredoxin-2 (Trx-2) is known as one of typical VDPs mainly distributed in mitochondria.^[1,2] As shown in Sup. Fig 3, we identified that the probe can selectively bind to the VDPs, indicating that it is readily to label the VDPs inside cells. The small molecule structure makes Mito-ESq-635 very easy to penetrate the cell membrane by transmembrane interaction and reach to mitochondria in short time (less than 5 minute) because of the plus charge and suitable hydrophilicity-hydrophobicity itself. Our image-based colocalization analysis (Supplementary Figure 1) also indicates that the MitoESQ-635 labels the mitochondria for a variety of cell species, with high R ratio with the commonly used Mito Tracker.

We have modified the description in the main text and supporting information accordingly.

[1] Spyrou G., Enmark E., Miranda-Vizuete A. and Gustafsson J. Cloning and expression of a novel mammalian thioredoxin. *J. Biol. Chem.* **272**, 1997: 2936-2941;
[2] Tanaka T., Hosoi F., Yamaguchi-Iwai Y., Nakamura H., Masutani H., Ueda S., Nishiyama A., Takeda S., Wada H., Spyrou G. and Yodoi J., Thioredoxin-2 (TRX-2) is an essential gene regulating mitochondria-dependent apoptosis, *The EMBO Journal*, **21**(7), 2002:1695-1703.

(3) I'm not sure that I agree with the authors that there is no phototoxicity. They describe changes in the mitochondria, 'bubble structures', with cristae growing inside, which I would interpret as swelling of the mitochondria. In particular, the changes in fig 3 between the first frame and the final frame look like swelling and fragmentation of the mitochondria, which is you would expect to see with phototoxicity. If this was the 'normal' state of mitochondria, it should be evident from the start of the imaging, not just later on.

Response:

We are sorry for our previous overstatement on non-phototoxicity. After careful investigation, we agree with the reviewer that there exists some phototoxicity during long time imaging process in Fig. 3. The case in Fig. 3 is a demonstration of light induced cell unhealthy, including swelling and fragmentation of the mitochondria, but with less significant fluorescence signal degradation.

During STED fluorescence nanoscopy, because of the power requirement for the donut beam, the balance between the phototoxicity (cell viability and organelle healthy) and the photodamage to the fluorescence dye is much more challenging. Lots of efforts have been spent to meet those issues, including improved STED super resolution techniques and novel dye developments [1-7]. Especially, it lacks the dyes of better performance for living cell STED imaging. During the review process of our manuscript, another group reported a photostable fluorescent marker for super resolution STED live imaging, termed MitoPB Yellow ($\lambda_{\text{ex}} = 488 \text{ nm}$; $\lambda_{\text{STED}} = 660$

nm), in which the swelling phenomenon is also observed [8].

Figure 3 STED imaging of the mitochondrial cristae, and the change of mitochondria width, fluorescence intensity, as well as image resolution. (a) Time-lapse STED imaging of the mitochondria. Scale bar, 3 μm . (b) The enlarge of red box in (a) and intensity profile at the red line. Scale bar, 2 μm . (c) Time-lapse plots of mitochondria width, fluorescence intensity, and image resolution. (d) Changes of mitochondria width and fluorescence intensity under different STED intensity.

For box plot of width of mitochondria, every frame has 6 data points, which means 6 mitochondria in the frames are analyzed. Their widths are detected by using intensity profile in ImageJ. The figure is plot using Origin software, Plot-Statistics-Box Chart. For fluorescence intensity, using the same original images data with width mitochondria, choose 6 squares (10×10 pixels) in the frames. In ImageJ, Images – Stacks – Measure Stacks, then the total intensity of every square (10×10 pixels) can be obtained in all frames. In every frame, we have 6 intensity data (6 squares), calculated the mean value intensity of every frame. For resolution, the intensity profile is obtained by using ImageJ, and Gaussian Cumulative Fit Peak is processed with Origin software.

Scale bar: 1 μm . 200 Frames, 10 Minutes

Supplementary Video 5 Drop1 KO HeLa with long mitochondria also have tendency to become bubbles, but with less speed. Dyed with 100 nM MitoESq-635 and with Prolong live antifade reagent (P36975, Thermo Fisher) for reducing photobleaching. 3s/frame, walking average, scale bar: 1 μm . 200 frames, 10 min. (a) Excitation 633 nm, 5 μW ; STED 775 nm, 8.96 mW. (b) Excitation 633nm, 5uW; STED 775nm, 7.84 mW. (c) Excitation 633 nm, 2.5 μW ; STED 775 nm, 6.72 mW.

- [1] Lanzano, L., et al. "Encoding and decoding spatio-temporal information for super-resolution microscopy." *Nature communications* 6.1 (2015): 1-9.
- [2] Tortarolo, G., et al. "Photon-separation to enhance the spatial resolution of pulsed STED microscopy." *Nanoscale* 11.4 (2019): 1754-1761.
- [3] Donnert, G., et al. "Macromolecular-scale resolution in biological fluorescence microscopy." *Proceedings of the National Academy of Sciences* 103.31 (2006): 11440-11445.
- [4] Laissue, P. P., et al. "Assessing phototoxicity in live fluorescence imaging." *Nature methods* 14.7 (2017): 657-661.
- [5] Marta Fernández-Suárez & Alice Y. Ting, Fluorescent probes for super-resolution imaging in living cells. *Nature Reviews Molecular Cell Biology*, 9, 929–943 (2008).
- [6] Hans Blom and Jerker Widengren, Stimulated Emission Depletion Microscopy. *Chemical Reviews* 2017, 117, 11, 7377-7427.
- [7] Gražvydas Lukinavičius, Gyuzel Y. Mitronova, Sebastian Schnorrenberg, Alexey N. Butkevich, Hannah Barthel, Vladimir N. Belov and Stefan W. Hell, Fluorescent dyes and probes for super-resolution microscopy of microtubules and tracheoles in living cells and tissues. *Chem. Sci.*, 2018,9, 3324-3334.
- [8] Wang, C., et al. "A photostable fluorescent marker for the super resolution live imaging of the dynamic structure of the mitochondrial cristae." *Proceedings of the National Academy of Sciences* 116.32 (2019): 15817-15822.

(4) This swelling also appears evident in Fig 4, as the width of the mitochondria appears to increase significantly over time, and not just due to fusion. While it appears that

they can in fact catch fusion events, due to the swelling, I'm not sure how well the cristae dynamics they observe reflect what normally happens during fusion.

Response:

We agree with the reviewer's comment. The swelling may influence the cristae dynamics together with other environmental conditions (such as temperature, pressure, pH, and so on). Here we tried to reduce this influence by using short frame time (3.9s per frame) and long dark recovery time (71.5 sec) to avoid light induced swelling.

In order to investigate the influence on the mitochondrial swelling induced by light exposure, we have added a new experiment to quantitatively analyse the width of the mitochondria over time, under different STED beam powers, as shown in the revised Figure 3. It shows the width and fluorescence intensity change faster when STED intensity is higher. The 775 nm STED power of 7-9 mW should be an optimized intensity in our case with relatively little photobleaching and photodamage. To further improve the resolution, STED intensity should be increased, in which case a longer dark recovery can help to reduce photobleaching and photodamage.

The long-term STED imaging of mitochondria can separate the phototoxicity and photodamage, enabling us to visualize photo-induced fusion events of mitochondria at high spatial resolution.

(5) In order to accurately compare the Mito-ESq-635 dye to other dyes used to image mitochondrial cristae, they should show their own data using their scope and settings, rather than pulling numbers from other publications, as settings and laser power are often unique to each microscope.

Response:

We agree with the reviewer. In figure 1, we compared our results with the results by EM from other publications which can be considered as ground truth of the shape of mitochondria cristae.

In order to accurately compare the Mito-ESq-635 dye to other dyes used to image mitochondrial cristae in live cells at molecule level resolution, we compared our results

with current published results (see supplementary note table 1). Our MitoESq-635 shows better performances over other reported STED compatible mitochondria probes.

Supplementary Table 1 Comparison of different results

Videos	Image time: t1 (s)	Rest time: t2 (s)	Total frame time: t3=t1+t2	Frames (F)	Video time: t4=t3*F (s)	Logical size	Pixel size (nm)	Excitation power	STED power (mW)	Resolution (nm)
SI Video 3	2.58	0.42	3s	60	180	512*512	28	5 μ W	7.84	70
SI Video 5a	2.58	0.42	3s	200	600	128*128	22	5 μ W	8.96	70
SI Video 5b	2.58	0.42	3s	200	600	128*128	31	5 μ W	7.84	80
SI Video 5c	2.58	0.42	3s	200	600	128*128	23	2.5 μ W	6.72	80
SI Video 7	2.58	0.42	3s	120	360	400*128	24	5 μ W	7.84	N/A
Scientific Report ^[1]	N/A	N/A	15s	10-20	120	N/A	20-25	N/A	N/A	70
PNAS ^[2]	N/A	N/A	1.3 s	300	390	N/A	N/A	N/A	108	60
Hessian SIM ^[3]	N/A	N/A	5.3 ms 188 HZ	800	4.2	N/A	N/A	66.15 W/cm ²	N/A	90

N/A: Not applicable

Note that the resolution in Figure 3a and Supplementary Video 5 are better than that of Hessian SIM, although SIM has a faster frame speed. Our results have longer video time than that of supplementary reference ^[1] and need less STED power which means less photodamage than supplementary reference ^[2], although their resolutions are similar.

[1] Stephan T., et al. Live-cell STED nanoscopy of mitochondrial cristae. *Scientific reports* **9**(1), **2019**: 1-6.

[2] Wang C., et al. A photostable fluorescent marker for the superresolution live imaging of the dynamic structure of the mitochondrial cristae. *Proceedings of the National Academy of Sciences* **116**(32), 2019: 15817-15822.

[3] Huang X., et al. Fast, long-term, super-resolution imaging with Hessian structured illumination microscopy. *Nature biotechnology* **36**(5), 2018: 451.

General comments:

(1) I would completely remove the first paragraph, which I feel is unnecessary. The evolutionary history of mitochondria and their role in cancer are not really relevant to the current work.

Response: Thanks for the suggestion. We have revised the first paragraph in the revision.

(2) Though mutations in OPA1 cause an optic atrophy phenotype, OPA1 does not encode a photoreceptor, and this has nothing to do with sensitivity to light or phototoxicity.

Response: Thanks for your kind suggestion. As also suggested by Reviewer 2, we have deleted the description of OPA1 in the revision.

(3) The authors need to include a bit more information regarding the deconvolution analysis they are applying to their images.

Response: Thank you for your suggestion. We use the Huygens Software for deconvolution analysis directly. Images are taken using Leica TCS SP8 STED 3X system, then uploaded to Huygens Software. Deconvolution process is completed by using auto setting in Huygens Software embedded in the system. Detailed parameters are:

Parameter	Setting
Background:	Automatic estimation
Estimate mode:	Lowest
Area radius:	0.7
deconvolution algorithm:	CMLE'
Maximum iteration:	40
Signal to noise ratio:	7
Quality threshold:	0.05
Iteration mode:	optimized
Bleaching correction:	If possible
PSFs per brick:	One PSF
Brick layout:	Auto

(4) Regarding the average widths of cristae, they use 42 nm and 43 nm for the width following fission. I assume this should be the same number.

Response: We apologize for our typo. It should be 42 nm and we have revised it in the revision.

(5) The authors state ‘the fluorescent molecule could melt’, with high laser power. Not sure that ‘melt’ is the correct term here.

Response: Thank you for your suggestion. Here we wish to express the phenomenon of burning-like signal bursting, as shown in Supplementary Figure 8. This is typically caused by excessive heat deposit due to the high STED laser power. We have hence changed “melting” into “bursting”.

Supplementary Figure 8. Bursting caused by high dose depletion beam power in STED microscopy. Depletion beam power, 51 mW. The molecules in red circle of STED image are bursting induced by high dose depletion beam. Scale bar, 5 μm .

The term “melting” is sometimes referring to the complex phenomenon in which both photo-induced organelle damage and burning-like signal bursting [1], such as the figure below:

[REDACTED]

Link: <https://www.abberior-instruments.com/products/expert-line/rescue/> (Accessed on Feb. 8, 2020)

(6) While the images in Sup Fig 2 certainly look like mitochondrial networks, it would be nice to see co-localization with another mitochondrial marker for these other cell lines.

Response: We agree that co-localization is helpful to illustrate correct labelling. Figure R1 is co-localization in U2OS, and figure R2 is co-localization in HeLa, figure R3 is co-localization in SH-SY5Y and SKOV cells.

(7) Generally speaking, many of the images are too small to adequately see at 100% magnification.

Response: Thanks for the suggestion. We have enlarged all the small images, such as Figure 4.

(8) The co-localization images in Sup fig 1 are not great.

Response: We have performed a new set of experiments with different cell lines, as shown in the Supplementary Figure 1. All the Pearson coefficients are >0.8 , indicating a highly colocalization between the Mito Tracker Green and our Mito-ESQ-635.

(9) For the supplementary videos 5 and 6, it would be nice to see the white dashed boxed areas shown in Fig 4 in order to focus the attention to the correct position.

Response: Thanks for the suggestion. We added the white dashed boxed in supplementary video 8 (video 6 in old version) in the revision as you can see below. The Supplementary video 5 is deleted because we revised Figure 3 and Figure 4 in main text.

SI Video 8 Raw data of Figure 4h: skeleton image of mitochondrial revealed cristae dynamics during fusion quantitatively. Scale bar, 5 μm .

Reviewer #2 (Remarks to the Author):

The manuscript by Yang and coworkers addresses a timely need in super-resolution imaging, namely the generation of live-cell imaging compatible probes to follow mitochondrial dynamics. The topic per se is thus relevant and interesting to a wide readership. A very similar paper appeared in Proceedings of the National Academy of Sciences earlier this year (Wang et al., PNAS 116, p15817 (2019)). In that paper, also a mitochondria-selective dye was presented for STED imaging, allowing observation of cristae dynamics. Therefore, claims of novelty such as “for the first time” in the abstract and elsewhere are problematic and that prior work should be acknowledged. Imaging in the far red as in the present manuscript (STED laser at 775 nm) is desirable to reduce interactions with cellular constituents. In terms of imaging performance, I would rate the PNAS paper higher (albeit at 660 nm STED wavelength) and I would also rate a recent paper by the Jakobs group higher in terms of imaging quality, using SNAP tag and silicon rhodamine for depletion at 775 nm (Stephan et al., Sci Rep 640920 (2019)). Therefore, the specific merit of the manuscript is to provide a label for STED microscopy at 775 nm which has affinity for mitochondria.

However, the manuscript has several shortcomings, such that I cannot recommend it for publication in its present state. Several of the claims are not sufficiently supported by the quality of the data. In other instances, the information provided is insufficient to truly assess the validity of the measurements.

Response:

Thanks for the reviewer’s recognition on the importance of this research and the specific merit of our new live cell Mitochondrial probe for STED nanoscopy. Indeed, a struggle against photodamage as well as photobleaching, blinking and saturation is always there if using fluorescence as contrast agents. Because of the power requirement for donut beam, the struggle against those flaws in STED fluorescence nanoscopy is much more challenging. Lots of efforts have been spent to meet those issues, including improved STED super resolution instruments and novel probes developments [1-7]. Especially, the dyes of better performance for living cell STED imaging are lacking. During the review process of our manuscript, as the reviewer 2 mentioned, Shigehiro Yamaguchi group and Stefan Jakobs group released their works with quite nice results. We have added the two references accordingly.

In our revision, we have performed several new experiments, and the images/ videos and quantitative analyses are updated.

For the MitoPB Yellow ($\lambda_{\text{ex}} = 488 \text{ nm}$; $\lambda_{\text{STED}} = 660 \text{ nm}$) is real a photostable fluorescent marker for super resolution STED live imaging [8], the STED power (108 mW-270 mW) in during their video recording is real high which is not so nice for cell imaging. Also, most of the images/videos shown in their paper are after deconvolution which will let the images looks much sharper and clearer but with artifacts. However, most of

the images and all the videos in Scientific report paper and our update manuscript are raw data.

For the COX8A-SNAP fusion proteins, real beautiful images and video shown in their paper but limited frame number/ imaging periods. Also, note that the OX8A-SNAP fusion proteins need quite complex operating steps. However, our probe offers the option of imaging mitochondria in living cells using STED nanoscopy without the necessity of an additional tagging step. Taking higher photostability and a lower saturation intensity, our MitoESq-635 probe revealed quite long term (200 frames, 600 s, see figure 3 and Supplementary Video 5) and high resolution (down to ~35 nm) live mitochondrial imaging under low level STED power. The supplementary table 2 compared our probes with the two side by side.

[1] Lanzano, L., et al. "Encoding and decoding spatio-temporal information for super-resolution microscopy." *Nature communications* 6.1 (2015): 1-9.

[2] Tortarolo, G., et al. "Photon-separation to enhance the spatial resolution of pulsed STED microscopy." *Nanoscale* 11.4 (2019): 1754-1761.

[3] Donnert, G., et al. "Macromolecular-scale resolution in biological fluorescence microscopy." *Proceedings of the National Academy of Sciences* 103.31 (2006): 11440-11445.

[4] Laissue, P. P., et al. "Assessing phototoxicity in live fluorescence imaging." *Nature methods* 14.7 (2017): 657-661.

[5] Marta Fernández-Suárez & Alice Y. Ting, Fluorescent probes for super-resolution imaging in living cells. *Nature Reviews Molecular Cell Biology*, 9, 929–943 (2008).

[6] Hans Blom and Jerker Widengren, Stimulated Emission Depletion Microscopy. *Chemical Reviews* 2017, 117, 11, 7377-7427.

[7] Gražvydas Lukinavičius, Gyuzel Y. Mitronova, Sebastian Schnorrenberg, Alexey N. Butkevich, Hannah Barthel, Vladimir N. Belov and Stefan W. Hell, Fluorescent dyes and probes for super-resolution microscopy of microtubules and tracheoles in living cells and tissues. *Chem. Sci.*, 2018,9, 3324-3334.

[8] Wang, C., et al. "A photostable fluorescent marker for the super resolution live imaging of the dynamic structure of the mitochondrial cristae." *Proceedings of the National Academy of Sciences* 116.32 (2019): 15817-15822.

Supplementary Table 2 Comparison of different probes

Probe	Molecule specificity	Wavelength	Frame time	Video time	Frames and time before swelling	Excitation power	STED power	Resolution	Data processing	Operational complexity
MitoESq-635	N/A	Excitation: 635 nm, STED: 775nm	2.58 s	600s, 200 frame s	No swelling for < 100 frames, 300s	5 μ W	7.84 mW	<50 nm	Raw data	One step
SNAP-tag ^[1]	COX8A-SNAP fusion proteins	Excitation: 640 nm, STED: 775nm	15 s	120s, 10-20 frame s	10-20 frames, 120s	N/A	N/A	70 nm	Raw data	Complex
Mito PB Yellow ^[2]	N/A	Excitation: 488 nm, STED: 660nm	1.3 s	390s, 300 frame s	About 100 frames, 130s	N/A	108 mW	60 nm	Deconvolution	N/A

N/A: Not applicable

In the meantime, a paper published on Beilstein Journal of Organic Chemistry cited our preprint on bioRxiv. Just like the recently published squaraine variant dye MitoESq-635 [58], our SiR dye 15 offers the option of imaging mitochondria in living cells using STED nanoscopy without the necessity of an additional tagging step. In contrast to MitoESq-635, our SiR dye 15 selectively stains mitochondria without background from unspecific membrane staining. However, higher photostability and a lower saturation intensity for STED result in a better performance in time-lapse live cell STED imaging of MitoESq-635. Taken together, our SiR dye 15 is a valid compromise between MitoESq-635 and SiR-Mito 8 offering nontoxic, specific mitochondrial staining in live cell STED imaging.

Matthias, Jessica, et al. "Synthesis of a dihalogenated pyridinyl silicon rhodamine for mitochondrial imaging by a halogen dance rearrangement." *Beilstein journal of organic chemistry* 15.1 (2019): 2333-2343.

In summary, we have listed several merits of our enhanced squaraine variant dye (MitoESq-635) for mitochondrial cristae live STED imaging:

- (1) The low saturation intensity and high photostability make it ideal for long-term, high-resolution STED nanoscopy.
- (2) MitoESq-635 is compatible with the live cell, with low photo-toxicity.
- (3) MitoESq-635 with a phenylarsenate moiety is able to covalently bind to VDPs, make sure the simple cell incubation.
- (4) Fluorescent enhancement upon binding to mitochondria, reducing background signals.

Please find my detailed assessment below.

Major points:

- (1) Absorption coefficient, quantum yield, fluorescence lifetime etc. for the squaraine dye should be given.

Response:

Thanks for the reviewer's suggestions. We have investigated the fluorescence quantum yields of MitoESq-635 and fluorescence lifetime in solution and live cells, respectively. Firstly, we measured the relative fluorescence quantum yield and molar extinction coefficient of Mito-ESq-635 in different solvent (see Table 1 and Table 2).

Absorption and Fluorescence spectra were measured with a UV/Vis absorption spectrometer (GBC Cintra 2020, Australia) and a fluorescence spectrometer (Horiba iHR320, American), respectively. The relative fluorescence quantum yields were determined with Rhodamine B as a standard and calculated using the following equation:

$$\Phi_x = \Phi_s \frac{I_x A_s \lambda_{exs}}{I_s A_x \lambda_{exs}} \left(\frac{n_x}{n_s} \right)^2$$

where A is absorbance at the excitation wavelength; I is the integration of the emission spectra; Φ_s represents the fluorescence quantum yield of the reference standard (Rhodamine B $\Phi_s = 0.97$ in ethanol solvents)^[2]; λ_{ex} is the excitation wavelength; n is the refractive index of the solution (because of the low concentrations of the solutions (10^{-7} - 10^{-8} mol/L), the refractive indices of the solutions were replaced with those of the solvents); and the subscripts x and s refer to the unknown and the standard, respectively.

Absorption coefficient: $\epsilon = A / [c]$, where A is absorbance at the excitation wavelength; [c] is the molar concentration.

Table 1 Fluorescence Quantum yield of Mito-ESq-635 and its relative parameters

	λ_x	A_x	F_x	η_x	Φ_x
Ethanol	610	0.10	1046.68	1.10	0.18
DCM	610	0.13	1133.13	1.42	0.26
DMSO	610	0.11	835.29	1.48	0.25

λ_x the excitation wavelength; A_x the absorbance of MitoESq-635 at the excitation wavelength; F_x the integral of the emission spectral curve of MitoESq-635; η_x the refraction index of solvents; Φ_x

fluorescence quantum yields of MitoESq-635

Note that the fluorescence quantum yield of MitoESq-635 is lower in very polar solvents (ethanol) than those in less polar solvents (DCM, DMSO), in particular, in water it is difficult to calculate the integral of the fluorescence (Fx), probably due to the high polarity of water. We tried to measure fluorescence lifetime of the probe in PBS solution instead.

Table 2 Molar Extinction coefficient of Mito-ESq-635

Solvents	DCM	DMSO	EtOH	MeOH	H ₂ O
$\epsilon(\times 10^5/\text{M}^{-1}\cdot\text{cm}^{-1})$	0.96	0.6	1.58	1.0	0.59

Supplementary Figure 14

Optical properties of MitoESq-635 in different solvents and living cells

(a-b) Absorption and Fluorescence spectra of MitoESq-635 in different solvents (c) Fluorescence lifetime decay curves of MitoESq-635 in different solvents (d-e) FLIM imaging of Mito-ESq-635 in HeLa cell and PBS which was measured by DCS-120 time-correlated single photon counting equipment (DCS-120, Becker Hickl), excitation wavelength: 635 nm; detect range: 675/30 nm. On the mitochondria of Hela cells, the lifetime $T \approx 1.7$ ns. Meanwhile, the lifetime of MitoESq-635 in PBS (1 μM) is $T_1 \approx 65$ ps.

The relative fluorescence quantum yields were measured on a Horiba fluorescent spectrometer with the method reported by H.H. Tønnesen [4]. And the fluorescence lifetime was measured on a commercially available equipment (DCS-120 time-correlated single photon counting equipment). Note that the fluorescence lifetime of MitoESq-635 is largely dependent on the environmental polarity. From the

fluorescence decay curves in different solvents (e.g. DMSO, ethanol, glycerol, water, PBS), the lifetime in DMSO or glycerol was calculated to be 1.65, 1.69 ns, but cannot be detected in very polar solvents like ethanol or water in supplementary Figure 12c. And fluorescence lifetime was further measured in PBS and living cells. From the results in supplementary figure 12d, 12e, it can be found the lifetime was markedly increased in living cells (1.7 ns), due to the large changes of microenvironments.

[4] Velapoldi, R.A. and H.H. Tønnesen, *Corrected Emission Spectra and Quantum Yields for a Series of Fluorescent Compounds in the Visible Spectral Region*. Journal of Fluorescence, 2004. **14**(4): p. 465-472.

(2) Reporting of imaging parameters and timing of recordings is insufficient, making it either impossible to gauge the true merits or rendering statements misleading. E.g. the section on STED imaging in the Methods section states that each frame was obtained at 1 s and the image series were obtained at a 1-minute interval but it is not clear to which measurements this refers. Precise timing information is warranted in each image caption and detailed imaging parameters should be stated for each measurement. Similarly, all other experiments should be described in sufficient detail to reproduce them.

Response: We agree with the reviewer the sufficient detail is needed to reproduce them. We have put all the parameters of images and videos in Supplementary table 1.

Supplementary Table 1 Comparison of different results

Videos	Image time: t1 (s)	Rest time: t2 (s)	Total frame time: t3=t1+t2	Frames (F)	Video time: t4=t3*F (s)	Logical size	Pixel size (nm)	Excitation power	STED power (mW)	Resolution (nm)
SI Video 3	2.58	0.42	3s	60	180	512*512	28	5 μ W	7.84	70
SI Video 5a	2.58	0.42	3s	200	600	128*128	22	5 μ W	8.96	70
SI Video 5b	2.58	0.42	3s	200	600	128*128	31	5 μ W	7.84	80
SI Video 5c	2.58	0.42	3s	200	600	128*128	23	2.5 μ W	6.72	80
SI Video 7	2.58	0.42	3s	120	360	400*128	24	5 μ W	7.84	N/A
Scientific Report ^[1]	N/A	N/A	15s	10-20	120	N/A	20-25	N/A	N/A	70
PNAS ^[2]	N/A	N/A	1.3 s	300	390	N/A	N/A	N	108	60
Hessian SIM ^[3]	N/A	N/A	5.3 ms 188 HZ	800	4.2	N/A	N/A	66.15 W/cm ²	N/A	90

N/A: Not applicable

[1] Stephan T., et al. Live-cell STED nanoscopy of mitochondrial cristae. *Scientific reports* **9**(1), **2019**: 1-6.

[2] Wang C., et al. A photostable fluorescent marker for the superresolution live imaging of the dynamic structure of the mitochondrial cristae. *Proceedings of the National Academy of Sciences* **116**(32), **2019**: 15817-15822.

[3] Huang X., et al. Fast, long-term, super-resolution imaging with Hessian structured

illumination microscopy. *Nature biotechnology* **36**(5), 2018: 451.

(3) In the abstract the authors state that live imaging over 50 minutes is possible. This is a meaningless statement if the actual time it takes for one frame and the intermittent (resting) times are not stated. As far as I understand, this statement refers to Suppl. Video 4, where each frame took about 4 seconds and then a recovery period of about 70 seconds was used, yielding 40 images in total. The authors should rather state the number of images that were taken.

Response: We agree with that the frame time, resting time and number of images are important. Here, we have performed a new experiment to image the Drop1 KO HeLa which has more healthy elongated mitochondria (as shown in Video R1). The frame time is 3s. Moreover, to investigate the parameters affecting the resolution, brightness, and the width of the mitochondria in living cells, we performed quantitative analysis on long-term live STED imaging of mitochondria (See figure 3, and Supplementary Video 5)

Scale bar: 1 μ m. 200 Frames, 10 Minutes

Supplementary Video 5 Drop1 KO HeLa with long mitochondria also have tendency to become bubbles, but with less speed. Dyed with 100 nM MitoESq-635 and with Prolong live antifade reagent (P36975, Thermo Fisher) for reducing photobleaching. 3s/frame, walking average, scale bar: 1 μ m. 200 frames, 10 min. (a) Excitation 633nm, 5 μ W; STED 775 nm, 8.96 mW. (b) Excitation 633 nm, 5 μ W; STED 775 nm, 7.84 mW. (c) Excitation 633 nm, 2.5 μ W; STED 775nm, 6.72 mW.

(4) The authors claim a resolution of 35.2 nm. This statement rests on a single line profile shown and a histogram without any detail on how data was analyzed. From the supplementary videos and the image material provided, I am very doubtful that signal-to-noise ratio is sufficient to claim 35 nm resolution or even 35.2 nm. The authors are encouraged to refer to the relationship of resolution and signal-to-noise ratio in STED microscopy, discussed e.g. by the Vicidomini group (Tortarolo et al., Evaluating image resolution in stimulated emission depletion microscopy. *Optica*, 5, p32 (2018)). Also the resolution determination in Fig. 2c is problematic due to insufficient signal-to-noise

ratio. From most of the images provided, I find it hard to clearly delineate individual cristae. In comparison, cristae stand out very clearly in the Scientific Reports paper from the Jakobs group.

Response: We thank the reviewer for bringing up the important literature, and the FRC method. Here we have used FRC-based resolution metrics to evaluate the resolution, as shown in Supplementary Figure 10. The FRC solution is $\sim 43\text{nm}$ which is close to intensity profile resolution 35.2nm .

Supplementary Figure 10.

FRC of Figure 4b. (a) Enlarge of figure 4b. 256×256 pixels, $16\mu\text{m}$ per pixel. (b) FRC of (a) shows the resolution is $\sim 43\text{nm}$, which is close to the resolution 35.2nm in Figure 4c (using intensity profile). The FRC is calculated with open software miplib [3].

[3] Koho, Sami, et al. "Fourier ring correlation simplifies image restoration in fluorescence microscopy." *Nature communications* 10.1 (2019): 1-9.

(5) I am somewhat concerned with the specificity of labeling for mitochondria. In Suppl. Fig. 1, there seems to be strong correlation with ER, which is, however, mentioned in the main text.

Response: We have performed a new set of experiments for the correlation between Mito-ESQ-635 and Mito Tracker. As can be seen from the Supplementary Figure 1, a Pearson Coefficient of $R > 0.8$ can be obtained for all the four different cell lines, indicating an excellent colocalization between our dye and mitochondria.

The previous result may be due to the overdosed concentration of the dye, which also targets the ER network.

(6) p5: The description of Suppl. Fig. 6 in the main text implies that no toxicity arises during 1 h of STED imaging. However, in Suppl. Fig. 6 it is not clear whether STED imaging has been applied in addition to the incubation. It is not described what the viability assay is.

Response: In Suppl. Fig. 6, we mainly checked the cytotoxicity of the probe itself to live cells, chemical toxicity of fluorescent probe. From the result, it can be found that the fluorescent probe (Mito-ESq635) is safe to live cells (almost no chemical toxicity to live cells) under our experimental conditions. We did not use the STED laser to illuminate the cell sample in this case.

(7) p 5: The statement on saturation intensity is contradictory. This measurement is not described properly. From the data provided it is not clear that this measurement was done in a way that excludes e.g. the possibility that bleaching might influence the measurement outcome. While superior imaging performance over Atto647N would be an achievement, this statement would need to be supported by further data.

Response: We apologize for our clerical error. Here we changed the original text 'higher' into 'lower'.

Original: 'Yet, the squaraine-STED dye presents a lower saturation intensity, 0.893 mW, which is ~3.4-fold higher than that of ATTO 647N (Figure 1).'

Revised: 'Yet, the squaraine-STED dye presents a lower saturation intensity, 0.893 mW, which is ~3.4-fold lower than that of ATTO 647N (Figure 1).'

We agree that the measurement may be influenced by bleaching. In Figure 1, when STED power is small, the effect of bleaching is small. When STED power becomes bigger, bleaching may influence fluorescence intensity. To be honest we didn't exclude the possibility that bleaching might influence the measurement outcome. But we believe these data can reflect the relationship between the saturation intensity of the squaraine-STED dye and ATTO 647N that squaraine-STED dye's saturation intensity is lower than ATTO 647N's.

(8) p5: I find the wording on 3D imaging somewhat misleading. The authors apply only resolution enhancement in the focal plane, not along the optical axis. Taking a stack of 5 STED images with xy resolution enhancement and confocal resolution axially is not particularly challenging. Therefore, also the statement in the conclusion "STED imaging of 3D stacks can reveal the ultrastructure of mitochondria in live cells" is not justified for the measurements presented in the manuscript.

Response: We agree that we apply only resolution enhancement in the focal plane, not along the optical axis. This is because we need to balance between xy resolution and optical resolution when fixing the power of STED beam. Comparing with the recent publications of STED imaging of mitochondria in live cell, we are currently the only

group that can demonstrate STED in a 3D z-stack, benefitted from the photostability of our Mito-ESQ-635 dye. We have made the following revision:

Original text: “STED imaging of 3D stacks can reveal the ultrastructure of mitochondria in live cells”.

Revised text: “STED imaging of 3D z-stacks can reveal the ultrastructure and spatial organization of mitochondria in live cells in different depths.”

(9) p7: The authors observe formation of bubbles and fission of mitochondria in long-term STED imaging. It should be checked that these dynamics are not related to the photo-burden inflicted by the imaging.

Response: We agree with the reviewer that there exists some phototoxicity during long time imaging process in Fig. 3. The case in Fig. 3 is a demonstration of light induced cell unhealthy, including swelling and fragmentation of the mitochondria, but with less significant fluorescence signal degradation.

During STED fluorescence nanoscopy, because of the power requirement for the donut beam, the balance between the phototoxicity (cell viability and organelle healthy) and the photodamage to the fluorescence dye is much more challenging. Lots of efforts have been spent to meet those issues, including improved STED super resolution techniques and novel dye developments [1-7]. Especially, it lacks the dyes of better performance for living cell STED imaging. During the review process of our manuscript, another group reported a photostable fluorescent marker for super resolution STED live imaging, termed MitoPB Yellow ($\lambda_{\text{ex}} = 488 \text{ nm}$; $\lambda_{\text{STED}} = 660 \text{ nm}$), in which the swelling phenomenon is also observed [8]. In the meantime, we developed an enhanced squaraine variant dye (MitoESq-635), to study the dynamic structures of mitochondrial cristae in live cells. The low saturation intensity and high photostability make MitoESq-635 ideal for long-term, high-resolution STED nanoscopy for live cell imaging.

In order to investigate the parameters affecting the resolution, brightness, the width of the mitochondria in living cells, we performed quantitative analysis on long term live STED imaging of mitochondria (See figure 3, and Supplementary Video 5)

Figure 3 STED imaging of the mitochondrial cristae, and the change of mitochondria width, fluorescence intensity, as well as image resolution. (a) Time-lapse STED imaging of the mitochondria. Scale bar, 3 μm . (b) The enlarge of red box in (a) and intensity profile at the red line. Scale bar, 2 μm . (c) Time-lapse plots of mitochondria width, fluorescence intensity, and image resolution. (d) Changes of mitochondria width and fluorescence intensity under different STED intensity.

For box plot of width of mitochondria, every frame has 6 data points, which means 6 mitochondria in the frames are analyzed. Their widths are detected by using intensity profile in ImageJ. The figure is plot using Origin software, Plot-Statistics-Box Chart. For fluorescence intensity, using the same original images data with width mitochondria,

choose 6 squares (10×10 pixels) in the frames. In ImageJ, Images – Stacks – Measure Stacks, then the total intensity of every square (10×10 pixels) can be obtained in all frames. In every frame, we have 6 intensity data (6 squares), calculated the mean value intensity of every frame. For resolution, the intensity profile is obtained by using ImageJ, and Gaussian Cumulative Fit Peak is processed with Origin software.

Scale bar: 1 μm . 200 Frames, 10 Minutes

Supplementary Video 5 Drop1 KO HeLa with long mitochondria also have tendency to become bubbles, but with less speed. Dyed with 100 nM MitoESq-635 and with Prolong live antifade reagent (P36975, Thermo Fisher) for reducing photobleaching. 3 s/frame, walking average, scale bar: 1 μm . 200 frames, 10 min. (a) Excitation 633 nm, 5 μW ; STED 775 nm, 8.96 mW. (b) Excitation 633 nm, 5 μW ; STED 775 nm, 7.84 mW. (c) Excitation 633 nm, 2.5 μW ; STED 775nm, 6.72 mW.

- [1] Lanzano, L., et al. "Encoding and decoding spatio-temporal information for super-resolution microscopy." *Nature communications* 6.1 (2015): 1-9.
- [2] Tortarolo, G., et al. "Photon-separation to enhance the spatial resolution of pulsed STED microscopy." *Nanoscale* 11.4 (2019): 1754-1761.
- [3] Donnert, G., et al. "Macromolecular-scale resolution in biological fluorescence microscopy." *Proceedings of the National Academy of Sciences* 103.31 (2006): 11440-11445.
- [4] Laissue, P. P., et al. "Assessing phototoxicity in live fluorescence imaging." *Nature methods* 14.7 (2017): 657-661.
- [5] Marta Fernández-Suárez & Alice Y. Ting, Fluorescent probes for super-resolution imaging in living cells. *Nature Reviews Molecular Cell Biology*, 9, 929–943 (2008).
- [6] Hans Blom and Jerker Widengren, Stimulated Emission Depletion Microscopy. *Chemical Reviews* 2017, 117, 11, 7377-7427.
- [7] Gražvydas Lukinavičius, Gyuzel Y. Mitronova, Sebastian Schnorrenberg, Alexey N. Butkevich, Hannah Barthel, Vladimir N. Belov and Stefan W. Hell, Fluorescent dyes and probes for super-resolution microscopy of microtubules and tracheoles in living cells and tissues. *Chem. Sci.*, 2018,9, 3324-3334.

[8] Wang, C., et al. "A photostable fluorescent marker for the super resolution live imaging of the dynamic structure of the mitochondrial cristae." *Proceedings of the National Academy of Sciences* 116.32 (2019): 15817-15822.

(10) Suppl. Note 3: No data is shown to back up the claims that the squaraine dye has wide applicability.

Response: We agree that this statement should be supported by further data. As the fluorescent MitoESq-635 reported in this work is a kind of new fluorophores derived from Squaraine dyes, rare work has been reported on the same fluorophore, especially used for super-resolution imaging. However, there are several works on traditional Squaraine dyes and its derivatives applied in biological imaging listed as follows:

(1) I-Che Wu, Jiangbo Yu, Fangmao Ye, Yu Rong, Maria Elena Gallina, Bryant S. Fujimoto, Yong Zhang, Yang-Hsiang Chan, Wei Sun, Xing-Hua Zhou, Changfeng Wu and Daniel T. Chiu, Squaraine-Based Polymer Dots with Narrow, Bright Near-Infrared Fluorescence for Biological Applications. *J. Am. Chem. Soc.* 2015, 137, 173–178

(2) Defan Yao, Yanshu Wang, Rongfeng Zou, Kexin Bian, Pei Liu, Shuzhan Shen, Weitao Yang, Bingbo Zhang, and Dengbin Wang, Molecular Engineered Squaraine Nanoprobe for NIR-II/Photoacoustic Imaging and Photothermal Therapy of Metastatic Breast Cancer. *ACS Appl. Mater. Interfaces* 2020, DOI: 10.1021/acsami.9b20147

(3) Kristina Ilina, William M. MacCuaig, Matthew Laramie, Jannatun N. Jeouty, Lacey R. McNally and Maged Henary, Squaraine Dyes: Molecular Design for Different Applications and Remaining Challenges. *Bioconjugate Chem.* 2019, DOI: 10.1021/acs.bioconjchem.9b00482

We have added the references in our revision.

(11) The Supplements, in particular Suppl. note 4 seem to be an unedited draft version with a series of problematic statements. For example, the statement on acquisition speed is not tenable in this unspecified way in view of the work by Schneider et al., *Nature Methods* 12, p827 (2015) where 125 frames per second super-resolution imaging rate was achieved.

Response: Thanks for your reminding and apologize for the problematic statements. We revised the Suppl. Note 4 by deleting the statement “Also, the acquisition speed of these methods (fourth column) is far slower than the biological dynamics in living cells, so it is difficult to get motion-induced artifacts, which means no single parameter can be optimized without compromising the others to technical consideration.”

(12) Suppl. Fig. 5: If MitoESq-635 is covalently linked to inner membrane proteins, it is no surprise that it stays bound also with prolonged imaging. This does, however, not imply that the mitochondrial membrane potential is still intact.

Response: We thank the reviewer for the constructive question. To address this, we performed a new colocalization experiment with the probe and Mitotracker green. The cell was fixed with 4% PFA after incubation of SKOV cells with the probe. As showed in Figure R1, the probe showed well colocalization imaging with the Mitotracker Green,

which means the probe covalently bound to the mitochondrial membrane VDPs. In the previous experiment, we used membrane potential dependent Mitotracker (Rho123) to colocalize with the probe in live cells, the mitochondrial membrane potential did not change a lot as the Rho123 showed well mitochondrial imaging. With scanning of strong laser, it can be clearly found that Rho123 slowly diffused out of mitochondria due to the change of the membrane potential, whereas MitoESQ-635 is still retained on mitochondria without marked diffusion. These results proved that the covalent binding of the probe with mitochondria, and the change in the mitochondrial membrane potential by the probe is much smaller than that of Rho 123 (Figure R2).

Figure R1 Co-localization experiment in fixed SKOV cells employing Mito Tracker Green as golden standard for mitochondrial marker; Cells were incubated with dye then fixed with 4% PFA; the first column, enhanced squaraine dye (20 nM) labeled fixed SKOV cells; the second column, Mito Tracker Green(50nM) labeled fixed SKOV cells; the third column, merged images; the fourth column, 2D intensity histogram and Pearson's coefficients; excitation wavelength: Mito ESq-635 (640 nm), Mito Tracker Green (488 nm); detect range: MitoESq-635 (700/75 nm nm), Mito Tracker Green (525/50 nm); Scale bar, 10 µm.

Figure R2 Time-course experiment on HeLa cells labeled with Rhodamine 123 or our Mito-ESQ635 dye for membrane potential (binding stiffness) evaluation. Here, Rhodamine 123 serves as golden standard for mitochondrial marker. The first row, Rhodamin 123 (50 nM) labeled HeLa cells; the second row, MitoESQ-635(20nM) labeled HeLa cells; excitation wavelength: Mito ESq-635 (635 nm), Rhodamine (488 nm); detect range: MitoESQ-635 (650-750 nm), Mito Tracker Green (500-600 nm); Scale bar, 10 μ m. The bottom chart represents the Intensity decay of Rhodamine 123 and MitoESQ-635, respectively.

Minor points:

1) p2: Wording is unfortunate: “ATP, which is the only energy form that cells can absorb”. ATP is certainly the most prevalent energy carrier, but it’s not absorbed by the cells but rather synthesized from ADP. Also there are other high-energy bonds in cells that can be used to drive biochemical processes.

Response: We apologize for the wrong statements here. We agree with that ATP is not absorbed by the cells and other high-energy bonds in cells can be used to drive biochemical process. As the first reviewer point out that first paragraph is relevant to the current work, we have deleted it in the revised txt.

2) p 2: I am not sure that mitochondrial dynamics “regulate” cancer cell activity.

Response: We thank the reviewer for pointing this out. We have revised the text.

Original text: “Its plasticity and structural dynamics can regulate a series of cancerous cell functions such as proliferation, migration, and resistance to therapy.”

Revised text: “Their various structural characteristics can reflect a number of cell activities, such as proliferation, migration, and resistance to therapy.”

3) p2: The statement on the sampling theorem is unfortunate. Resolution is defined as the capability to discern two nearby structures.

Response: We agree that resolution is defined as the capability to discern two nearby structures. To be honest, according to the sampling theorem, the pixel step should smaller than 50 nm to achieve 100nm resolution. We have revised the statement on the sampling theorem in the revised version.

Origin: “Considering the Nyquist-Shannon sampling theorem, one needs ~50 nm resolution to visualize the gaps between mitochondria cristae.”

Revised: “, which means imaging with resolution far below 100 nm is necessary to visualize the gaps between mitochondria cristae.”

4) p2: There is an inconsistency with the author name in the citation Dannie et al.

Response: Thanks for reminding. We corrected the author name in the revised version.

Origin: “Dannie et al. also demonstrated label-free third-order photoacoustic (PA) nanoscopy of the cytochromes in the inner mitochondrial membrane in fibroblasts with 88 nm resolution.”

Revised: “Danielli et al. demonstrated label-free third-order photoacoustic (PA) nanoscopy of the cytochromes in the inner mitochondrial membrane in fibroblasts at 88 nm resolution.”

5) Several of the citations are incomplete.

Response: Thanks for reminding. We revised the citations in the revised version.

Old version:

21. Frezza, C. *et al.* OPA1 controls apoptotic cristae remodeling independently from mitochondrial fusion. **126**, 177-189 (2006).
25. Masullo, L. A. *et al.* Enhanced photon collection enables four dimensional fluorescence nanoscopy of living systems. **9**, 3281 (2018).
26. Stephan, T., Roesch, A., Riedel, D. & Jakobs, S. J. b. Live-cell STED microscopy of mitochondrial cristae. 640920 (2019).
35. Vogel, F., Bornhövd, C., Neupert, W. & Reichert, A. S. Dynamic subcompartmentalization of the mitochondrial inner membrane. *J Cell Biol* **175**, 237-247 (2006).
36. van de Linde, S., Sauer, M. & Heilemann, M. Subdiffraction-resolution fluorescence imaging of proteins in the mitochondrial inner membrane with photoswitchable fluorophores. *Journal of structural biology* **164**, 250-254 (2008).
37. Lea, P. J. & Hollenberg, M. J. Mitochondrial structure revealed by high-resolution scanning electron microscopy. *American journal of anatomy* **184**, 245-257 (1989).

New Version:

17. Frezza, C. *et al.* OPA1 controls apoptotic cristae remodeling independently from mitochondrial fusion. *Cell* **126**, 177-189 (2006).
21. Masullo, L. A. *et al.* Enhanced photon collection enables four dimensional fluorescence nanoscopy of living systems. *Nature Communications* **9**, 3281 (2018).
22. Stephan, T., Roesch, A., Riedel, D. & Jakobs, S. J. b. Live-cell STED microscopy of mitochondrial cristae. *Scientific Reports* **9**, 12419(2019).
29. Vogel, F., Bornhövd, C., Neupert, W. & Reichert, A. S. Dynamic subcompartmentalization of the mitochondrial inner membrane. *Journal of Cell Biology* **175**, 237-247 (2006).
30. van de Linde, S., Sauer, M. & Heilemann, M. Subdiffraction-resolution fluorescence imaging of proteins in the mitochondrial inner membrane with photoswitchable fluorophores. *Journal of Structural Biology* **164**, 250-254

(2008).

- 31 Lea, P. J. & Hollenberg, M. J. Mitochondrial structure revealed by high-resolution scanning electron microscopy. *American Journal of Anatomy* **184**, 245-257 (1989).

6) p3: It would be helpful to include a statement on the absorption wavelength range of OPA1.

Response: Thanks for reminding. Based on the first reviewer's comment, we have deleted the description of OPA1.

7) p3: "Stimulated emission" is not a laser process. "Laser" means "light amplification by stimulated emission of radiation". Stimulated emission is an important component of the laser process, but not all stimulated emission is equivalent to lasing.

Response: We agree with the reviewer. We have deleted this description "(laser process)" for clarity.

Original: "the fluorescence in the "donut" area should be converted to stimulated emission (laser process) through high power laser illumination"

Revised: "the fluorescence in the "donut" area should be converted to stimulated emission through high power laser illumination"

8) p3: "...the efficiency is generally too low to generate effective STED processing." This sentence needs rewording.

Response: We thank the reviewer for pointing this out. We have revised it in the revised text.

Original: "they are not photostable enough to endure long-term STED imaging, and the efficiency is generally too low to generate effective STED process"

Revised: "they are not photostable enough to endure long-term STED imaging at high resolution".

9) p4: I am not sure whether the authors want to state that they label the "plasma membrane".

Response: Yes, it is exactly what we mean. We agree with that there is some correlation with ER when the dye is in high density. We hope to use this one dye to see both mitochondria and ER at the same time in living cell. We want to distinguish them by using a program. To be honest we are still research on it. One idea is distinguishing them by different intensity information between them. In other words, mitochondria are brighter than ER, we want to use this information to distinguish them. Another idea we are thinking about is using machine learning to distinguish them when we have enough training data.

10) Fig 1c: The authors plot power, not “intensity”, as stated in the figure caption.

Response: Sorry for our wrong words. We have revised the word ‘power’ in the figure into ‘intensity’ in the revised version.

[REDACTED]

Figure 1 Development of a highly photostable and minimally phototoxic bright cell-permeable enhanced squaraine probe for mitochondria inner membrane labeling. (a) Chemical structure of MitoESq-635 used for the specific labeling of VDPs. (b) Absorption and emission spectrum of the enhanced squaraine dye, for which a 775 nm pulse laser can be employed for depletion and a 635 nm laser for excitation during STED setup. (c) The detected fluorescence signal as a function of the depletion beam intensity; the excitation beam was 635 nm with a 5 ps pulse width and 80 MHz, and the STED beam was 775 nm with 600 ps pulse width and 80 MHz.

11) In Suppl. Fig. 1, the last column is not adequately described.

Response: Thank you for your reminder. We added the description about Suppl. Fig 1 in the new version: “the last column is 2D intensity histogram, and R is Pearson’s R value. This analysis is performed with Fiji (ImageJ) plugin Coloc 2. The results show that for all the cell lines, the Pearson colocalization coefficients are greater than 0.8.”

12) p 8: What do the authors precisely mean by “melting” of dye molecules?

Response: Thank you for your suggestion. Here we wish to express the phenomenon of burning-like signal bursting, as shown in Supplementary Figure 8. This is typically caused by excessive heat deposit due to the high STED laser power. We have hence changed “melting” into “bursting”.

Supplementary Figure 8. Bursting caused by high dose depletion beam power in STED microscopy. Depletion beam power, 51 mW. The molecules in red circle of STED image are bursting induced by high dose depletion beam. Scale bar, 5 μm .

The term “melting” is sometimes referring to the complex phenomenon in which both photo-induced organelle damage and burning-like signal bursting [1], such as the figure below:

[REDACTED]

Link: <https://www.abberior-instruments.com/products/expert-line/rescue/> (Accessed on Feb. 8, 2020)

13) Fig. 4: What is a “skeletal image”?

Response: Thanks for suggestion. We added explain for skeletal image in Methods and materials. You can also find it as below.

Skeleton connectivity quantitative analysis is similar to our previous work^[32] Skeletonize3D (download from <http://imagej.net/Skeletonize3D>), a Fiji plugin, was utilized to perform the skeletonization of mitochondria, and their geometrical and topological features of the original structure were obtained^[33]. Thereby the lengths and numbers of mitochondrial structural skeletons were counted for the morphological binary images.

14) The data in Suppl. Fig. 4 is consistent with covalent labeling, but not a “proof”.

Response: Thanks for reminding. We agree that Suppl. Fig. 4 is consistent with covalent labeling, but not a “proof”. We have revised it in the revised text.

Original: “Proof of covalent binding of MitoESq-635 to the VDPs inside cells;”

Revised: “Consistent with covalent binding of MitoESq-635 to the VDPs inside cells;”

15) Suppl. note 1, line 66: “literature procedure” would warrant a citation.

Response: We agree that a citation is needed. And we have added the citation in the revised version.

Citation:

1 Huang, C., Yin, Q., Zhu, W., Yang, Y., Wang, X., Qian, X., & Xu, Y. (2011). Highly Selective Fluorescent Probe for Vicinal - Dithiol - Containing Proteins and In Situ Imaging in Living Cells. *Angewandte Chemie International Edition*, 50(33), 7551-7556.

Reviewer #3 (Remarks to the Author):

The paper titled “Mitochondrial dynamics quantitatively revealed by STED nanoscopy with an enhanced squaraine variant probe” reported a new fluorescence living cell probe, MitoESq-635. It is an excellent candidate for live cell STED nanoscopy, which will make real impacts on pushing super resolution microscopy towards practical live cell STED imaging. The manuscript is well written and timely. With much better performance over the existing mitochondrial probes, the authors obtained beautiful super resolution images of mitochondrial ultrastructure in 3D and long term (about 1 hour) using STED nanoscopy. They got mitochondrial cristae dynamical videos with a decent resolution of sub-50 nm for 1 hour for the first time in super resolution microscopy and mitochondrial research field. I have several minor questions for the authors to address before it is published on Nature Communications.

1. In figure 1b, I suggest the authors to add the labels of excitation and STED wavelengths.

Response: Thanks for your reminding. We added them in figure 1b in the revised version.

[REDACTED]

Figure 1 Development of a highly photostable and minimally phototoxic bright cell-permeable enhanced squaraine probe for mitochondria inner membrane labeling. (a) Chemical structure of MitoESq-635 used for the specific labeling of VDPs. (b) Absorption and emission spectrum of the enhanced squaraine dye, for which a 775 nm pulse laser can be employed for depletion and a 635 nm laser for excitation during STED setup. (c) The detected fluorescence signal as a function of the depletion beam intensity; the excitation beam was 635 nm with a 5 ps pulse width and 80 MHz, and the STED beam was 775 nm with 600 ps pulse width and 80 MHz.

2. In figure 2c, figure 3c and figure 4d, the details of how the data points were fitted should be described.

Response: We have added descriptions of the details data processing in Figure 3 and Figure 4 in Methods and materials: Data processing.

In Figure 3, for box plot of width of mitochondria, every frame has 6 data points, which means 6 mitochondria in the frames are analyzed. Their widths are detected by using intensity profile in Fiji. The figure is plot using Origin software, Plot-Statistics-Box Chart. For fluorescence intensity, using the same original images data with width mitochondria, choose 6 squares (10 pixels x 10 pixels) in the frames. In Fiji, Images – Stacks – Measure Stacks, then the total intensity of every square (10 pixels x 10 pixels) can be obtained in all frames. In every frame, we have 6 intensity data (6 squares), calculated the mean value intensity of every frame. To normalize the intensity, all intensities are divided by the mean value of the intensity first frame. Then calculated the mean value and variance of intensity in every frame by using Statistics on Rows in Origin software. At last plot Y errors. For resolution, the intensity is obtained by using Fiji, and Gaussian Cumulative Fit Peak is processed with Origin software. In Figure 4, data fitting is finished by using Gaussian Cumulative Fit Peak tool in Origin software.

3. In Supplementary Figure 9, all the scale bars on fluorescence images were missing. **Response:** Sorry for the missing scale bar. We have added them in the revision. You can also find it as below.

[REDACTED]

Supplementary Figure 9.

The comparison of ultra-fine structure of mitochondria were resolved by STED and EM [1,2]. Scale bar, black 500 nm, white, 1 μm .

4. Although this probe worked pretty well with STED nanoscopy, the performance of this probe in live cell STED could be further improved if it can be combined with recent new STED technologies such as RESCUE STED, MINFIELD, DyMIN, STEDD, et al. Can the authors discuss about this?

Response: We agree that combined with new STED technologies such as RESCUE STED, MINFIELD, DyMIN, STEDD is helpful in living cell image.

For RESCUE STED, excitation and STED are shut off at positions where no signal arises. In conventional STED these places are pre-stressed and pre-recorded. Thus, RESCUE STED can reduce photobleaching in STED. Combined with RESCUE STED and this probe, should be beneficial to living cell image. [R1]

MINFIELD reduces bleaching in STED/RESOLFT nanoscopy through restricting the scanning to subdiffraction-sized regions. By safeguarding the molecules from the intensity of the maxima and exposing them only to the lower intensities (around the minima) needed for the off-switching (on-switching), MINFIELD largely avoids detrimental transitions to higher molecular states. A bleaching reduction by up to 100-fold is demonstrated. Combined with MINFIELD STED and this probe, bleaching can be reduced even more. [R2, R3]

The principle of DyMIN is that it only uses as much on/off-switching light as needed to image at the desired resolution. Fluorescence can be recorded at those positions where fluorophores are found within a subresolution neighborhood. DyMIN is shown to lower the dose of STED light on the scanned region up to ~20-fold under common biological imaging conditions, and >100-fold for sparser 2D and 3D samples. Combined with MINFIELD STED and this probe, bleaching can be reduced even more as well. [R4]

Stimulated emission double depletion (STEDD) as a method to selectively remove artificial background intensity. This method is beneficial when considering lower-power, less redshifted depletion pulses. By combining STEED and this probe, we can keep balance in both better resolution and less bleaching for longer imaging. This probe is also beneficial to STEED.

To sum up, combined with those new STED technologies above with this probe is helpful in living cell image.

[R1] Thorsten Staudt, Andreas Engler, Eva Rittweger, Benjamin Harke, Johann Engelhardt, and Stefan W. Hell, "Far-field optical nanoscopy with reduced number of state transition cycles," *Opt. Express* 19, 5644-5657 (2011)

[R2] F. Göttfert, T. Pleiner, J. Heine*, V. Westphal, D. Görlich, S. J. Sahl, S. W. Hell, "Strong signal increase in STED fluorescence microscopy by imaging regions of

subdiffraction extent", PNAS 1621495114 (2017)

[R3] J. Heine, M. Reuss, B. Harke, E. D'Este, S. J. Sahl and S. W. Hell, "Adaptive-illumination STED nanoscopy", PNAS (2017)

[R4] Nature review on DyMIN STED: G. Pacchioni, "Always look on the bright side of the fluorophore", Nature Review Materials (2017)

Reviewers' comments:

Reviewer #1 (Remarks to the Author):

Overall, the data and images in the manuscript show that the MitoESq-635 dye is a valuable tool for imaging mitochondria via STED microscopy. Yang et al. have made several key changes that help improve the manuscript. However, while the response to the reviews was quite thorough, I did not feel the revised version completely addressed all of the key points discussed. Moreover, in my opinion some of the claims in the paper are still overstated. For example, there is clearly photobleaching which occurs over time, and there is clearly phototoxicity with the mitochondrial morphology and cristae structure changing over time. This will likely be true for any dye and does not detract from the paper, but to claim otherwise is misleading. What is valuable is the direct side by side comparison between the new dye and existing options, which clearly demonstrate the superiority of MitoESq-635 (e.g. Sup fig 7). Perhaps some of this data could be moved to data in the main paper rather than supplemental.

Specific Issues:

Additional details for the imaging over 50 minutes are required in the abstract.

Be careful about the specificity of various statements. The following is not true:

“conventional optical microscopy techniques are insufficient to visualize the structure and dynamics of mitochondria”

Generally speaking, researchers have been using conventional microscopy for nearly two decades to study mitochondrial structure and dynamics. However, if you are referring to sub-mitochondrial structures (cristae) being imaged live, the sentence would be more appropriate.

Not sure how this sentence fits in the context of a paragraph on microscopy

“The mitochondrial intermembrane space can be measured with indirect approaches such as proteomic mapping”

The following statement is still false: “Mitochondria are more sensitive to light than other cellular organelles due to the existence of photoreceptors”

Mitochondria do not have photoreceptors. Photoreceptors are a cell type, which have mitochondria. Furthermore, this is irrelevant to mitochondrial sensitivity to light.

The text is still unclear with reference to the dye labeling the plasma membrane. The plasma membrane is the membrane at the exterior of the cell and is distinct from the mitochondria and mitochondrial membranes. Thus, it is unclear how the dye is binding to the plasma membrane and labelling mitochondria. Perhaps the authors mean the dye can bind membranes generally, rather than the plasma membrane specifically?

The following sentence is poorly constructed: “Benefitted from its low saturation intensity, the MitoESq635 probe exhibited no marked toxicity in HeLa cells at a concentration of 1 μ M after 1 hour of incubation during STED superresolution imaging microscopy”. However, the bigger issue is that details on the imaging conditions are still absent, making the statement useless.

While the following statement does have some key information, it is still incomplete: “With MitoESq-635, rare photobleaching and mitochondrial shape variations are observed upon exposure to STED scanning for over 10 minutes (1.2 s per frame, 700 \times 700 pixels, 12.6 μ m \times 12.6 μ m, STED beam of 30.2 mW at 775 nm), as shown in Supplementary Figure 7”. The reader needs to know how many images were captured over the 10 minutes. Is this one image every minute (e.g. a 1 minute recovery), or one image every 1.2 seconds (e.g. constant imaging).

In the context of the Z-axis, how relevant are steps of 200nm, especially given that cristae are 30-50 nm in width?

Please clarify what the red and green colors in Fig 4i represent.

For sup fig 7, be consistent with the labeling of MitoESq-635. In one panel it is the blue data points on the graph, and in the next panel it is the orange data points. Can these graphs be combined? Again, details matter, there are different details given for the MitoEs1-635 vs Mitotracker Green, than for the MitoESq-635 vs Mitotracker FarRed (e.g. dye concentrations). Why are the concentrations different for MitoESq-635 and Mitotracker FarRed, is this still a valid comparison?

There are still several small typos/formatting issues throughout that required correction. For example, "standard gold live cell" should be "gold standard live cell"

Reviewer #2 (Remarks to the Author):

The authors have made improvements to the manuscript. However, they have glossed over several of the reviewers' comments and have given explanations to reviewers without fully addressing the actual issues in the text. The paper is still not exact enough in many aspects, especially concerning imaging. I am still reluctant to recommend publication.

Please find some specific points below.

- 1) The authors have not removed some of the statements that I find problematic from the abstract and elsewhere, such as 35.2 nm resolution and "for long periods of time with minimal phototoxicity." In fact, phototoxicity seems to be a problem, as mitochondria do display marked swelling. The authors now quantify swelling of mitochondria. Statements such as "showed healthy behavior with few changed to spheroidicity" might still be perceived as downplaying potential phototoxicity, especially as in the discussion, the other super-resolution imaging options for mitochondria are called out as being too phototoxic. Also in the present paper there seems to be a substantial light effect on the mitochondria and it should be discussed more fully that much of the observed alterations/dynamics could simply be due to phototoxicity. It should also be noted that phototoxicity is not just dependent on laser power. A large portion of it is mediated by fluorophore bleaching, such that the bleaching rate and dye concentration are also important parameters for phototoxic responses.
- 2) The paper by Wang et al. in PNAS should be acknowledged in the main text and the specific merits of the two approaches discussed. It is not sufficient to cite it as a supplementary reference.
- 3) I pointed out in my previous comments that signal-to-noise ratio is an important factor when determining resolution. E.g. the resolution measurement in Frame 60 of Fig. 3 is not meaningful. Signal-to-noise ratio in Fig. 2c is too low to perform a reliable resolution measurement on it.
- 4) Fig. 1c: On what sample is the saturation intensity measured? How is such a measurement taken precisely? The authors state in the rebuttal letter that they cannot exclude that bleaching contributes to the apparently reduced saturation intensity of their probe vs. Atto 647N. Without a proper description of the procedures, it is not clear what the merit of such a measurement is and whether the conclusions drawn are valid.
- 5) Reviewer #1 asked for a description of deconvolution. It is not helpful if this is in the response to the reviewer and does not enter the methods section.
- 6) "Plasma membrane" is the membrane that separates the cell interior from the outside. From the authors' explanation in the rebuttal letter I am still not sure they want to state that they label the plasma membrane rather than a mitochondrial membrane.
- 7) Power and intensity in Fig.1 are still mixed up. Milliwatts are not a unit for intensity. This also applies to the saturation intensities stated.
- 8) Can the authors please state how they assess viability (Suppl. Fig. 8, see my comment in the original referee report)? The data in Suppl. Fig. 8 is meaningless if the authors don't give the

details on how they perform such an analysis.

9) The new section on data processing is not understandable.

10) The caption in suppl. fig. 9 and 10 does not explain how cells were labelled. It would be good to have such information directly in each figure caption.

11) Suppl. Fig. 14: I cannot read the axes in panels d and e.

12) Whenever a nonlinear lookup table is given, a colour bar for the lookup table should be provided. This also applies to the 2D histograms in suppl. figs. 1-3.

13) Suppl. Fig. 2: It appears that slightly different focus positions are imaged in the two colour channels, lowering overlap.

14) Hemozoin is a crystalline substance, such that "melting" is more appropriate there than for dye molecules.

Reviewer #3 (Remarks to the Author):

After revision, I recommend this manuscript to be published.

Response Letter

Manuscript ID: NCOMMS-19-16887A

Title: Mitochondrial dynamics quantitatively revealed by STED nanoscopy with an enhanced squaraine variant probe

Reviewer #1

Response to comment

Comment: *Overall, the data and images in the manuscript show that the MitoESq-635 dye is a valuable tool for imaging mitochondria via STED microscopy. Yang et al. have made several key changes that help improve the manuscript. However, while the response the reviews was quite thorough, I did not feel the revised version completely addressed all of the key points discussed. Moreover, in my opinion some of the claims in the paper are still overstated. For example, there is clearly photobleaching which occurs over time, and there is clearly phototoxicity with the mitochondrial morphology and cristae structure changing over time. This will likely be true for any dye and does not detract from the paper, but to claim otherwise is misleading. What is valuable is the direct side by side comparison between the new dye and existing options, which clearly demonstrate the superiority of MitoEsq-635 (e.g. Sup fig 7). Perhaps some of this data could be moved to data in the main paper rather than supplemental.*

Response: Thank you for your helpful review and your recognition on several key improvements in the revised manuscript. We agree that some of claims in the paper are still overstated such as phototoxicity. We revised the overstated claims in the revised manuscript to make it more rigorous (you can find the changes as follow). We also move the comparison figure between the new dye and existing options (old SI figure 7) to main text to highlight the superiority of MitoEsq-635. We are appreciated for your kind suggestions again.

Changes about overstated claims are as follow:

Changes:

Abstract, Paragraph 1, (Page 1)

*“Our study demonstrates the emerging capability of optical STED nanoscopy to investigate intracellular physiological processes at nanoscale resolution for long periods of time **with minimal phototoxicity.**”*

Changed to:

“Our study demonstrates the emerging capability of optical STED nanoscopy to investigate intracellular physiological processes at nanoscale resolution for long period of time.”

Changes:

Introduction, Paragraph 4, (Page 3)

“To address the challenges in long term, high resolution STED live cell imaging, in this work, we have developed a new squaraine dye derivative (MitoESq-635) that is compatible with live cells and produces low phototoxicity.”

Changed to:

“To address the challenges in long term STED live cell nanoscopic imaging, in this work, we have developed a new squaraine dye derivative (MitoESq-635) that is compatible with live cells.”

Changes:

Results, Photostability and stimulated emission saturation intensity of the dye, Paragraph 3 (Page 7)

“Benefitted from its low saturation intensity, the MitoESq-635 probe exhibited no marked toxicity in HeLa cells at a concentration of 1 μ M after 1 hour of incubation during STED superresolution imaging microscopy (Supplementary Figure 8). The low saturation intensity, extended photostability, and low toxicity make the dye very suitable for long-term STED imaging in live cells.”

Changed to:

“The MitoESq-635 probe exhibited low toxicity in HeLa cells at a concentration of 1 μ M after 1 hour of incubation (Supplementary Figure 9), which suggests that the morphological changes of mitochondria are primarily from the STED imaging light dose. But with the concentration and incubation time increase, cytotoxicity of the probe will happen as well. The low saturation intensity, and extended photostability make the dye very suitable for long-term STED imaging in live cells.”

Changes:

Results, Subcellular dynamic nanoscopic imaging with MitoESq-635, (Page 9)

“Taking advantage of the live cell compatibility of MitoESq-635, the rapid nanoscale spatiotemporal mitochondrial dynamics in a living HeLa cells was captured with STED (1.2 s per frame, 700 \times 700 pixels, 12.6 μ m \times 12.6 μ m, STED beam of 33.6 mW at 775 nm before the objective) long-term (over 10 minutes) (Supplementary Video 4), and (3s per frame, exposure time 2.58s, rest for 0.42s, STED power at 8.96mW,

7.84mW and 6.72mW, 200 frames, 10 minutes) (Supplementary Video 5), during which the mitochondria showed healthy behavior with few changed to spheroidicity.”

Changed to:

“Taking advantage of the live cell compatibility of MitoESq-635, the rapid nanoscale spatiotemporal mitochondrial dynamics in a living HeLa cells was captured with STED (3s per frame, exposure time 2.58s, rest for 0.42s, STED power at 8.96 mW, 7.84 mW and 6.72 mW, 200 frames, 10 minutes) (Supplementary Video 1).”

Here is the response to specific issues one by one.

Specific Issues:

Response to comment 1

Comment: *Additional details for the imaging over 50 minutes are required in the abstract.*

Response: Thank you for your suggestions. We agree that details for the imaging over 50 minutes are required in the abstract. We add them in the revised text as follows.

Changes:

Abstract, Paragraph 1, (Page 1)

“We demonstrate the time-lapsed imaging of the mitochondrial inner membrane over 50 minutes in living HeLa cells at 35.2 nm resolution for the first time.”

Changed to:

“We demonstrate the time-lapsed imaging of the mitochondrial inner membrane over 50 minutes (3.9s per frame, and with 71.5s dark recovery) in living HeLa cells at 35.2 nm resolution for the first time.”

Response to comment 2

Comment: *Be careful about the specificity of various statements. The following is not true: “conventional optical microscopy techniques are insufficient to visualize the structure and dynamics of mitochondria” Generally speaking, researchers have been using conventional microscopy for nearly two decades to study mitochondrial structure and dynamics. However, if you are referring to sub-mitochondrial structures (cristae) being imaged live, the sentence would be more appropriate.*

Response: Thank you for the reviewer’s expertise. We fully agree that it is more accurate to use “conventional optical microscopy techniques are insufficient to visualize the structure and dynamics of mitochondria”. We have revised it in the text.

Changes:

Introduction, Paragraph 1, (Page 2)

“Yet, limited by the diffraction of light, conventional optical microscopy techniques are insufficient to visualize the structure and dynamics of mitochondria.”

Changed to:

“Yet, limited by the diffraction of light, conventional optical microscopy techniques are insufficient to visualize **the sub-mitochondrial structures (cristae) and dynamics of these sub-mitochondrial structures** ².”

Added reference:

2 **Jakobs, S. & Wurm, C. A. Super-resolution microscopy of mitochondria. *Current Opinion in Chemical Biology* 20, 9-15 (2014).**

Response to comment 3

Comment: *Not sure how this sentence fits in the context of a paragraph on microscopy: “The mitochondrial intermembrane space can be measured with indirect approaches such as proteomic mapping”*

Response: Thank you for your suggestions. Originally, we would like to discuss there are some approaches other than optical microscopy to study on sub-mitochondrial structures. We agree with your comment here, and this sentence has been deleted for consistency of paragraph 1 and paragraph 2 in revised manuscript.

Delete:

Introduction, Paragraph 1, (Page 2)

The mitochondrial intermembrane space can be measured with indirect approaches such as proteomic mapping.

Response to comment 4

Comment: *The following statement is still false: “Mitochondria are more sensitive to light than other cellular organelles due to the existence of photoreceptors” Mitochondria do not have photoreceptors. Photoreceptors are a cell type, which have mitochondria. Furthermore, this is irrelevant to mitochondrial sensitivity to light.*

Response: Thank you for your comment and apologize for our fault here. We change the statement in the revised manuscript.

Changes:

Introduction, Paragraph 3, (Page 3)

*“Mitochondria are more sensitive to light than other cellular organelles **due to the existence of photoreceptors**; excessive light exposure can cause mitochondrial dysfunction and mitophagy.”*

Changed to:

“Mitochondria are more sensitive to light than other cellular organelles; excessive light exposure can cause mitochondrial dysfunction and mitophagy^[14-16].”

Added reference:

14 Trotta, A. P. & Chipuk, J. E. Mitochondrial dynamics as regulators of cancer biology. *Cellular and Molecular Life Sciences* 74, 1999-2017 (2017).

15 Burté, F., Carelli, V., Chinnery, P. F. & Yu-Wai-Man, P. Disturbed mitochondrial dynamics and neurodegenerative disorders. *Nature reviews neurology* 11, 11 (2015).

16 Frezza, C. et al. OPA1 controls apoptotic cristae remodeling independently from mitochondrial fusion. 126, 177-189 (2006).

Response to comment 5

Comment: *The text is still unclear with reference to the dye labeling the plasma membrane. The plasma membrane is the membrane at the exterior of the cell and is distinct from the mitochondria and mitochondrial membranes. Thus, it is unclear how the dye is binding to the plasma membrane and labelling mitochondria. Perhaps the authors mean the dye can bind membranes generally, rather than the plasma membrane specifically?*

Response: Thank you for your comment. Sorry for our misleading by using “plasma membrane”. Originally, we would like to mention the possibility for wide applications in fluorescent labeling of enhanced squaraine dyes derivatives (Supplementary Note 2). MitoESq-635 is one of enhanced squaraine dyes derivatives we designed for mitochondria. According to the experiment of colocalization (Supplementary Figure 1, 2, 3), MitoESq-635 is specially binding to the membrane proteins in the mitochondria. Thus, to make it clear, we revised these phrases in the revised manuscript.

Changes:

Results, Imaging live cells with the modified squaraine dye, Paragraph 1, (Page 4)

*“A hexylamidophenylarsenate chain is conjugated to the sulfide atom at the central position of the four-membrane ring in the squaraine dye (Supplementary Note 1), which can be used as a protein label for the **plasma membrane**. Due to the fast binding of phenylarsenate to vicinal dithiols and the prioritized targeting of mitochondria by the dye molecules, incubating live cells with MitoESq-635 for a few minutes is enough to label the **plasma membrane** with high density.”*

Changed to:

A hexylamidophenylarsenate moiety is conjugated to the sulfide atom at the central position of the four-membrane ring in the squaraine dye (Supplementary Note 1), which **can be potentially used as a protein label for the mitochondrial membrane or other organelles in live cells (Supplementary Note 2)**. Due to the fast binding of phenylarsenate to vicinal dithiols and the prioritized targeting of mitochondria by the dye

molecules, incubating live cells with MitoESq-635 for a few minutes is **sufficient** to label the **mitochondrial membrane** with high density.

Response to comment 6

Comment: *The following sentence is poorly constructed: “Benefitted from its low saturation intensity, the MitoESq-635 probe exhibited no marked toxicity in HeLa cells at a concentration of 1 μ M after 1 hour of incubation during STED superresolution imaging microscopy”. However, the bigger issue is that details on the imaging conditions are still absent, making the statement useless.*

Response: Thank you for your suggestions. As the results showed in Supplementary Figure 9, we mainly considered the cytotoxicity of the probe itself (Chemotoxicity) to live HeLa cells prior to the study of the phototoxicity of the probe to mitochondria. There is no STED image here, sorry for our mistake. It can be seen from Supplementary Figure 9 that the probe shows low cytotoxicity under low concentration of the probe and short incubation time. And with enhancement of probe concentration and elongation of the incubation time, the cytotoxicity of the probe gradually increased. Thus, we **added a new sentence “But when concentration and incubation time increase, toxicity will increase as well.”** to make it more rigorous. Here the calculation of cell viability is **added in revised Supplementary Figure 9.**

Changes:

Result, Photostability and stimulated emission saturation intensity of the dye, Paragraph 3, (Page 7)

Benefitted from its low saturation intensity, the MitoESq-635 probe exhibited low toxicity in HeLa cells at a concentration of 1 μ M after 1 hour of incubation **during STED superresolution imaging microscopy** (Supplementary Figure 8).

Changed to:

“The MitoESq-635 probe exhibited low toxicity in HeLa cells at a concentration of 1 μ M after 1 hour of incubation (Supplementary Figure 9), **which suggests that the morphological changes of mitochondria are primarily from the STED imaging light dose. But with the concentration and incubation time increase, cytotoxicity of the probe will happen as well.**”

Added:

Supplementary Figure 9, how we calculate cell viability

In this experiment, a commercial chemical reagent of CCK-8, being nonradioactive, allows sensitive colorimetric assays for the determination of the number of viable cells in cell proliferation and cytotoxicity assays. Herein, we use the products from Dojindo (CCK-8) to carry out the cytotoxic assay. The detailed experimental procedures are listed as follows:

100 μ L of HeLa cell suspension (about 5000 cells/well) was firstly dispensed in a 96-well plate, and was pre-incubated for 24 hours in a humidified incubator (37°C, 95% humidity, 5% CO₂). Then 10 μ L of various concentrations of MitoESq-635 (final concentrations are 0, 0.5, 1, 2, 5 μ M) were added to the 96-well plate to be tested, which the 96-well plate was incubated for different time (1, 3, 6 hours) in the incubator. And then 10 μ L of CCK-8 solution was carefully added to each well of the above plate to avoid introducing bubbles into the wells. The obtained 96-well plate was incubated for another 3 hours in the incubator. And finally, the absorbance of the samples at 450 nm were measured using a microplate reader (Rayto RT-6100, Shenzhen, China).

$$\text{Cell Viability} = [(A_s - A_b) / (A_c - A_b)] \times 100\%$$

where A_s is absorbance of experimental wells (containing medium, CCK-8, MitoESq-635); A_c represents the absorbance of control wells (containing medium, CCK-8); A_b stands for the absorbance of blank wells (containing CCK-8).

Response to comment 7

Comment: *While the following statement does have some key information, it is still incomplete: “With MitoESq-635, rare photobleaching and mitochondrial shape variations are observed upon exposure to STED scanning for over 10 minutes (1.2 s per frame, 700 × 700 pixels, 12.6 μ m × 12.6 μ m, STED beam of 30.2 mW at 775 nm), as shown in Supplementary Figure 7”. The reader needs to know how many images were captured over the 10 minutes. Is this one image every minute (e.g. a 1 minute recovery), or one image every 1.2 seconds (e.g. constant imaging).*

Response: Thank you for your suggestions. Here “(1.2 s per frame, 700 × 700 pixels, 12.6 μ m × 12.6 μ m, STED beam of 30.2 mW at 775 nm)” is the information of old Supplementary Video 4 in NCOMMS-19-

16887A. You can also find this information in section “Subcellular dynamic nanoscopic imaging with MitoESq-635” about old Supplementary Video 4 in NCOMMS-19-16887A. We are sorry for misuse of information here. To make it more clearly, we delete original Supplementary Video 4 in revised version, and add right information for Figures 1d and 1e (which is the original Supplementary Figure 7).

Changes:

Results, Photostability and stimulated emission saturation intensity of the dye, Paragraph 2, (Page 6)

“With MitoESq-635, rare photobleaching and mitochondrial shape variations are observed upon exposure to STED scanning for over 10 minutes (1.2 s per frame, 700 × 700 pixels, 12.6 μm × 12.6 μm, STED beam of 30.2 mW at 775 nm), as shown in Supplementary Figure 7”.

Changed to:

“With MitoESq-635, rare photobleaching and mitochondrial shape variations are observed upon exposure to STED scanning for **over 100 seconds (1 s per frame, imaging time: 0.66s, recovery time: 0.34s, STED beam of 30.2 mW at 775 nm)**, as shown in **Figure 1e**. Under the same imaging conditions, in contrast, the fluorescence signal from MitoTracker Green dropped very quickly due to significant photobleaching (>70% in 100 scans, Figure 1e), making it unsuitable for long-term live cell STED imaging. Supplementary Figure 8 shows similar result that MitoESq-635 exhibits much more robust photostability than MitoTracker. **When STED power is reduced, it is achieved by using MitoESq-635, that STED imaging over 200 frames 10 minutes (3s per frame, imaging time: 2.58s, recovery time: 0.42s, STED beam of 8.96 mW at 775 nm) as shown in Supplementary Video 1 and Supplementary Note 5.”**

Delete:

Old Supplementary Video 4

Delete:

Results, Subcellular dynamic nanoscopic imaging with MitoESq-635, Paragraph 1, (Page 8)

Text about old Supplementary Video 4

(1.2s per frame, 700 × 700 pixels, 12.6 μm × 12.6 μm, STED beam of 33.6 mW at 775 nm before the objective) long-term (over 10 minutes) (Supplementary Video 4)

Response to comment 8

Comment: *In the context of the Z-axis, how relevant are steps of 200 nm, especially given that cristae are 30-50 nm in width?*

Response: Thank you for your comments. Here, we present 3D z-stack STED imaging (multiple layer 2D STED) to give more spatial information about the mitochondria, comparing with 2D STED imaging in other figures (one layer). Previously, it was challenging to perform mitochondria STED imaging in 3D z-stacks because of photobleaching during imaging. We admit that it is better to improve the resolution in z axis with smaller z-steps; and even with 3D-STED (the donut also applies on z axis). Restricted to the STED microscopy system, we cannot achieve this at the moment. We believe that a better z-resolution and a smaller z-step can be achieved with our MitoESQ-635 in future.

Response to comment 9

Comment: *Please clarify what the red and green colors in Fig 4i represent.*

Response: Thank you for your comments. Here, green color means original image, and red color means connective pixels of morphological binary image. We agree that discussion about skeleton image can easily lead to misunderstand, thus we delete Figures 4i and 4j which are related to skeleton image in revised manuscript.

Delete:

Figures 4i and 4j

Delete:

Text about Figures 4i and 4j

(i) skeleton image of mitochondrial revealed cristae dynamics during fusion quantitatively. (j) connective points change with time (k) cristae dynamics during from line shape to bubble.

Response to comment 10

Comment: For sup fig 7, be consistent with the labeling of MitoESq-635. In one panel it is the blue data points on the graph, and in the next panel it is the orange data points. Can these graphs be combined? Again, details matter, there are different details given for the MitoEs1-635 vs Mitotracker Green, than for the MitoESq-635 vs Mitotracker FarRed (e.g. dye concentrations). Why are the concentrations different for MitoESq-635 and Mitotracker FarRed, is this still a valid comparison?

Response: Thank you for your comments and we agree with your comments. We changed the color to combine them. We are sorry for the different details between the two comparison. We added all concentrations, and both are 0.1 μM . The concentration for MitoTracker Deep red here is 0.1 μM as well, it is a clerical error when changing the figure but we forgot to update the label. And we move these figures into Figure 1. And we added new experiments to cross-check our results, as shown in Supplementary Figure 8.

Revised Figure 1:

[REDACTED]

Figure 1 Development of a highly photostable bright cell-permeable enhanced squaraine probe for mitochondria inner membrane labeling. (a) Chemical structure of MitoESq-635 used for the specific labeling of VDPs. (b) Absorption and emission spectrum of the enhanced squaraine dye, for which a 775 nm pulse laser can be employed for depletion and a 635 nm laser for excitation during STED setup. (c) The detected fluorescence signal of MitoESq-635 and ATTO 647N solution dye pool on coverslip as a function of the depletion beam intensity; the excitation beam was 635 nm with a 5 ps pulse width and 80 MHz, and the STED beam was 775 nm with 600 ps pulse width and 80 MHz. (d) Comparison of the photostability under a confocal laser microscope of HeLa cells co-stained with MitoESq-635 (0.1 μ M) and MitoTracker Green (Rhodamine123) (0.1 μ M). Upper row, MitoESq-635, excited at 633 nm, and collected at 645-680 nm. Lower row, MitoTracker Green (Rhodamine123), excited at 488 nm, and collected at 500-560 nm. Confocal images are under the same imaging conditions, excitation under 1.97 μ W averaged power, 1 frame per second acquisition speed (imaging time: 0.66s, recovery time: 0.34s). Fluorescence signal of each image are plotted as a function of the recorded image number. Scale bar, 10 μ m. (e) Comparison of the photostability under a STED nanoscope of living HeLa cells stained with MitoESq-635 (0.1 μ M) and MitoTracker DeepRed (0.1 μ M), respectively. STED images are under the same imaging conditions, excitation under 1.1 μ W averaged power at 640 nm, STED beam of 30.2 mW average power at 775 nm, 1 frame per second acquisition speed (imaging time: 0.66s, recovery time: 0.34s). Fluorescence signal of each image are plotted as a function of the recorded image number. Scale bar, 2.5 μ m.

Added:

Supplementary Figure 8

Supplementary Figure 8

Comparison of the photostability with MitoTracker and MitoESq-635.

(a) Comparison of the photostability under a confocal laser microscope of HeLa cells co-stained with 0.1 μM MitoESq-635 and 0.1 μM MitoTracker Green (Rhodamine123). Upper row, MitoESq-635, excited at 633 nm, and collected at 645–680 nm. Lower row, MitoTracker Green (Rhodamine123), excited at 488 nm, and collected at 500–560 nm. Confocal images are under the same imaging conditions, excitation under 1.97 μW averaged power, 1 frame per second acquisition speed (imaging time: 0.66s, recovery time: 0.34s). Fluorescence signal of each image are plotted as a function of the recorded image number. Scale bar, 10 μm . (b) Comparison of the photostability under a STED nanoscope of living HeLa cells stained with MitoESq-635 (0.1 μM) and MitoTracker DeepRed (0.1 μM), respectively. STED images are under the same imaging conditions, excitation under 1.1 μW averaged power at 640 nm, STED beam of 36 mW average power at 775 nm, 1 frame per second acquisition speed (imaging time: 0.66s, recovery time: 0.34s). Fluorescence signal of each image are plotted as a function of the recorded image number. Scale bar, 2 μm .

Response to comment 11

Comment: *There are still several small typos/formatting issues throughout that required correction. For example, “standard gold live cell” should be “gold standard live cell”*

Response: Thank you for your careful proofreading. We check the whole manuscript again and changed some small typos/formatting issues in revised manuscript.

Changes:

Results. Photostability and stimulated emission saturation intensity of the dye. Paragraph 2, (Page 6)
standard gold live cell

Changed to:

gold standard live cell

Changes:

Figure 2 Caption. (Page 8)

*“The scale bars of the EM and STED fluorescence images in **d** are 500 nm and 1 μm .”*

Changed to:

*“The scale bars of the EM and STED fluorescence images in **(d)** are 500 nm and 1 μm .”*

Changes:

Results. Subcellular dynamic nanoscopic imaging with MitoESq-635. Paragraph 3. (Page 11)

*“We also observed the fusion process of mitochondria, as shown in **Figure 4f and h** (Supplementary Videos 7 and 8).”*

Changed to:

“We also observed the fusion process of mitochondria, as shown in **Figures 4f and 4h** (Supplementary **Videos 6 and 7**).”

Changes:

Results. Subcellular dynamic nanoscopic imaging with MitoESq-635. Paragraph 3. (Page 11)

*“in these figures both fusion and fission processes **occurring** simultaneously.”*

Changed to:

“in these figures both fusion and fission processes occur simultaneously.”

Reviewer #2

The authors have made improvements to the manuscript. However, they have glossed over several of the reviewers' comments and have given explanations to reviewers without fully addressing the actual issues in the text. The paper is still not exact enough in many aspects, especially concerning imaging. I am still reluctant to recommend publication.

Please find some specific points below.

Response to comment 1

Comment: *The authors have not removed some of the statements that I find problematic from the abstract and elsewhere, such as 35.2 nm resolution and “for long periods of time with minimal phototoxicity.” In fact, phototoxicity seems to be a problem, as mitochondria do display marked swelling. The authors now quantify swelling of mitochondria. Statements such as “showed healthy behavior with few changed to spheroidicity” might still be perceived as downplaying potential phototoxicity, especially as in the discussion, the other super-resolution imaging options for mitochondria are called out as being too phototoxic. Also in the present paper there seems to be a substantial light effect on the mitochondria and it should be discussed more fully that much of the observed alterations/dynamics could simply be due to phototoxicity. It should also be noted that phototoxicity is not just dependent on laser power. A large portion of it is mediated by fluorophore bleaching, such that the bleaching rate and dye concentration are also important parameters for phototoxic responses.*

Changes:

Abstract, Paragraph 1, (Page 1)

*“Our study demonstrates the emerging capability of optical STED nanoscopy to investigate intracellular physiological processes at nanoscale resolution for long periods of time **with minimal phototoxicity.**”*

Changed to:

“Our study demonstrates the emerging capability of optical STED nanoscopy to investigate intracellular physiological processes at nanoscale resolution for long period of time.”

Changes:

Introduction, Paragraph 4, (Page 3)

“To address the challenges in long term, high resolution STED live cell imaging, in this work, we have developed a new squaraine dye derivative (MitoESq-635) that is compatible with live cells and produces low phototoxicity.”

Changed to:

“To address the challenges in long term STED live cell nanoscopic imaging, in this work, we have developed a new squaraine dye derivative (MitoESq-635) that is compatible with live cells.”

Changes:

Results, Photostability and stimulated emission saturation intensity of the dye, Paragraph 3 (Page 7)

“Benefitted from its low saturation intensity, the MitoESq-635 probe exhibited no marked toxicity in HeLa cells at a concentration of 1 μ M after 1 hour of incubation during STED superresolution imaging microscopy (Supplementary Figure 8). The low saturation intensity, extended photostability, and low toxicity make the dye very suitable for long-term STED imaging in live cells.”

Changed to:

“The MitoESq-635 probe exhibited low toxicity in HeLa cells at a concentration of 1 μ M after 1 hour of incubation (Supplementary Figure 9), which suggests that the morphological changes of mitochondria are primarily from the STED imaging light dose. But with the concentration and incubation time increase, cytotoxicity of the probe will happen as well. The low saturation intensity, and extended photostability make the dye very suitable for long-term STED imaging in live cells.”

Changes:

Results, Subcellular dynamic nanoscopic imaging with MitoESq-635, (Page 9)

“Taking advantage of the live cell compatibility of MitoESq-635, the rapid nanoscale spatiotemporal mitochondrial dynamics in a living HeLa cells was captured with STED (1.2 s per frame, 700 \times 700 pixels, 12.6 μ m \times 12.6 μ m, STED beam of 33.6 mW at 775 nm before the objective) long-term (over 10 minutes) (Supplementary Video 4), and (3s per frame, exposure time 2.58s, rest for 0.42s, STED power at 8.96mW, 7.84mW and 6.72mW, 200 frames, 10 minutes) (Supplementary Video 5), during which the mitochondria showed healthy behavior with few changed to spheroidicity.”

Changed to:

“Taking advantage of the live cell compatibility of MitoESq-635, the rapid nanoscale spatiotemporal mitochondrial dynamics in a living HeLa cells was captured with STED (3s per frame, exposure time 2.58s, rest for 0.42s, STED power at 8.96 mW, 7.84 mW and 6.72 mW, 200 frames, 10 minutes) (Supplementary Video 1).”

Response to comment 2

Comment: *The paper by Wang et al. in PNAS should be acknowledged in the main text and the specific merits of the two approaches discussed. It is not sufficient to cite it as a supplementary reference.*

Response: Thank you for your suggestions. We added the paper by Wang et al. in PNAS and the discussion of specific merits of the two approaches in the main text. You can find them in conclusion part, the same as follow.

Added:

Conclusion. (Page 13)

“Very recently, during this manuscript under review process, Jakobs group reported a new cell line expressing mitochondrial protein fused to SNAP-tag, to enable high resolution STED of mitochondrial cristae in live cell for 2 minutes (every 15s per frame, 8 frames)³². Wang et al. developed a new fluorescent labeling reagent and captured the structures of the mitochondrial cristae with a resolution of ~60 nm when depleted at 660 nm for 390s³³. In the two works, Jakobs group achieved high resolution (70 nm) STED of mitochondrial cristae in live cell for 2 minutes, and Wang et al. captured the structure of mitochondrial cristae with a resolution of ~60 nm (after deconvolution) for 390s (300 frames). Comparing with Ref. [32], our result has much longer imaging time. For Ref. [33], we obtained a better resolution with less STED power. A Detailed comparison can be found in Supplementary Tables 3 and 4 (Supplementary Note 5).”

Response to comment 3

Comment: *I pointed out in my previous comments that signal-to-noise ratio is an important factor when determining resolution. E.g. the resolution measurement in Frame 60 of Fig. 3 is not meaningful. Signal-to-noise ratio in Fig. 2c is too low to perform a reliable resolution measurement on it.*

Response: Thank you for your comments. We agree that signal-to-noise ratio is an important factor when determining resolution. Figure 3b also confirm this, when frame number is bigger, we can see the SNR is slower and resolution becomes worse. For Fig. 2c, here we have used FRC-based resolution metrics to evaluate the resolution, as shown in Supplementary Figure 10. The FRC solution is 59.1 nm which is close to intensity profile resolution 51 nm. We also added it into Supplementary Figure 10 as below.

The illumination and donut beams induce photobleaching as the imaging time increase. The time lapse STED images of living cell will have both fluorescence signal and resolution loss. What we want to show here is that the resolution will drop due to SNR and other complex factors in (figure 3b and c). As the reviewer pointed out here, the resolution measurement in Frame 60 is not so reliable due to its low SNR. We also put the FRC result panels here and add a discussion about SNR issues.

Added:

Results, Subcellular dynamic nanoscopic imaging with MitoESq-635, Paragraph 1, (Page 9)

“The time lapse STED images of living cell will have both fluorescence signal and resolution loss with frame number increasing. Note that due to its low SNR, the resolution measurement in Frame 60 is difficult to have a reliable cross section profile and determine the resolution relying on its FWHM, so we also put the FRC analysis results of frame #1 and #60 here.”

Added SI Figure:

Supplementary Figure 10

FRC of Figure 2c. (a) White block shows Figure 2c image. (b) FRC shows the resolution of (a) is 59.1 nm, which is close to the resolution in Figure 2c (using intensity profile). The FRC is calculated by open software miplib.

Revised Figure 3:

Figure 2 STED imaging of the mitochondrial cristae, and the change of mitochondria width, fluorescence intensity, as well as image resolution. (a) Time-lapse STED imaging of the mitochondria. Scale bar, 3 μm . (b) The enlarge of red box in (a), cross section intensity profile at the red line and their

FRC analysis results on resolution²⁶. Scale bar, 2 μm . (c) Time-lapse plots of mitochondria width, fluorescence intensity, and image resolution. (d) Changes of mitochondria width and fluorescence intensity under different STED intensity.

Response to comment 4

Comment: *Fig. 1c: On what sample is the saturation intensity measured? How is such a measurement taken precisely? The authors state in the rebuttal letter that they cannot exclude that bleaching contributes to the apparently reduced saturation intensity of their probe vs. Atto 647N. Without a proper description of the procedures, it is not clear what the merit of such a measurement is and whether the conclusions drawn are valid.*

Response:

Thank you for your suggestions. For materials with less photobleaching, such as quantum dots, nanoparticles are used to measure saturation intensity [R1]. For materials which can be influenced by photobleaching and dipole orientation, saturation measured by single molecule and thin film are used and compared by S. W. Hell, samples in solution are discussed as well [R2]. In our experiment, we first prepared for the dye solution (1 mM) by dissolving the sample in DMSO solvent. Then the obtained solution was diluted with ethanol solvent to make the experimental solution ($<0.1 \mu\text{M}$). Small amount of the diluted dye solution in ethanol ($<0.1 \mu\text{M}$) were dispersed onto a clean glass slide for the measurement of the STED saturation intensity using a commercial Leica-SP8 microscope. After the evaporation of the ethanol solvent, the residue of dye molecules on the glass slide was observed to be sparsely fluorescent spots, subsequently sealed with a coverslip. Some fluorescent particles were selected to measure the saturated intensity. The bleaching effect is reduced, although we can't avoid photobleaching completely. And with this method, we calculated the saturation intensity of ATTO 647N in our experiment to be $15 \text{ MW}/\text{cm}^2$, which is close to the data of $10 \text{ MW}/\text{cm}^2$ in reference [R3]. Therefore, we believed the saturation intensity of MitoESq-635 is valid as well.

[R1] Ye, Shuai, et al. "Low - Saturation - Intensity, High - Photostability, and High - Resolution STED Nanoscopy Assisted by CsPbBr₃ Quantum Dots." *Advanced Materials* 30.23 (2018): 1800167.

[R2] Westphal, V., et al. "Laser-diode-stimulated emission depletion microscopy." *Applied Physics Letters* 82.18 (2003): 3125-3127.

[R3] Bianchini, P., Harke, B., Galiani, S., Vicidomini, G., & Diaspro, A. (2012). Single-wavelength two-photon excitation–stimulated emission depletion (SW2PE-STED) superresolution imaging. *Proceedings of the National Academy of Sciences*, 109(17), 6390-6393.

Changes:

Results, Photostability and stimulated emission saturation intensity of the dye, Paragraph 3, (Page 6)

However, the squaraine-STED dye has a lower saturation intensity at 0.893 mW, which is ~3.4-fold lower than that of ATTO 647N (3.069 mW, Figure 1c).

Changed to:

However, the squaraine-STED dye has a saturation intensity of **4.37 MW/cm²**, which is ~3.4-fold lower than that of ATTO 647N (**15.0 MW/cm²**, Figure 1c, which is close to the data **10 MW/cm²** in reference²⁴).

Added reference:

- 24 Bianchini, P., Harke, B., Galiani, S., Vicidomini, G. & Diaspro, A. Single-wavelength two-photon excitation–stimulated emission depletion (SW2PE-STED) superresolution imaging. *Proceedings of the National Academy of Sciences* **109**, 6390, doi:10.1073/pnas.1119129109 (2012).

Response to comment 5

Comment: Reviewer #1 asked for a description of deconvolution. It is not helpful if this is in the response to the reviewer and does not enter the methods section.

Response: Thank you for your suggestions. We added it to the methods section in revised manuscript.

Added:

Methods and materials, Deconvolution, Page 16

“Deconvolution was performed by using the Huygens Software embedded in the Leica TCS SP8 STED 3X system. Deconvolution process is completed by using auto setting in Huygens. Detailed parameters are as follows: (1) Background: automatic estimation. (2) Estimate mode: lowest. (3) Area radius: 0.7. (4) Deconvolution algorithm: CMLE. (5) Maximum iteration: 40. (6) Signal to noise ratio: 7. (7) Quality threshold: 0.05. (8) Iteration mode: optimized. (9) Bleaching correction: if possible. (10) PSFs per brick: one PSF. (11) Brick layout: Auto.”

Response to comment 6

Comment: *“Plasma membrane” is the membrane that separates the cell interior from the outside. From the authors’ explanation in the rebuttal letter I am still not sure they want to state that they label the plasma membrane rather than a mitochondrial membrane.*

Response: Thank you for your comment. Sorry for our misleading by using “plasma membrane”. Originally, we would like to mention the possibility for wide applications in fluorescent labeling of enhanced squaraine dyes derivatives (Supplementary Note 2). MitoESq-635 is one of enhanced squaraine dyes derivatives we designed for mitochondria. According to the experiment of colocalization (Supplementary Figure 1, 2, 3), MitoESq-635 is specially binding to the membrane proteins in the mitochondria. Thus, to make it clear, we revised these phrases in the revised manuscript.

Changes:

Results, Imaging live cells with the modified squaraine dye, Paragraph 1, (Page 4)

*“A hexylamidophenylarsenate chain is conjugated to the sulfide atom at the central position of the four-membrane ring in the squaraine dye (Supplementary Note 1), which can be used as a protein label for the **plasma membrane**. Due to the fast binding of phenylarsenate to vicinal dithiols and the prioritized targeting of mitochondria by the dye molecules, incubating live cells with MitoESq-635 for a few minutes is enough to label the **plasma membrane** with high density.”*

Changed to:

A hexylamidophenylarsenate moiety is conjugated to the sulfide atom at the central position of the four-membrane ring in the squaraine dye (Supplementary Note 1), which **can be potentially used as a protein label for the mitochondrial membrane or other organelles in live cells (Supplementary Note 2)**. Due to the fast binding of phenylarsenate to vicinal dithiols and the prioritized targeting of mitochondria by the dye molecules, incubating live cells with MitoESq-635 for a few minutes is **sufficient** to label the **mitochondrial membrane** with high density.

Response to comment 7

Comment: *Power and intensity in Fig.1 are still mixed up. Milliwatts are not a unit for intensity. This also applies to the saturation intensities stated.*

Response: Thank you for your comments and sorry for our misuse. Refer to the reference [R1] below, now we add STED power, saturated power, a pulse peak intensity, and a pulse peak saturated intensity to make it clear. The relationship between STED power and a pulse peak intensity is presented as follow.

In the experiment, 775 nm STED is 600 ps pulse width, 80 MHz. Thus, pulse duty factor is $(1/(80 \text{ MHz}))/600 \text{ ps} = 20.83$.

At 775 nm, an oil immersion lens of NA = 1.4 yields $\sim 4.26 \cdot 10^{-9} \text{ cm}^2$. (In [R2], at 650 nm, an oil immersion lens of NA = 1.4 yields $\sim 3 \cdot 10^{-9} \text{ cm}^2$. Thus, for 775 nm it is about $3 \cdot 10^{-9} \text{ cm}^2 \cdot (775/650)^2 = 4.26 \cdot 10^{-9} \text{ cm}^2$)

Now we can calculate that 1 mW STED power is equal to $(1 \text{ mW}/4.26 \cdot 10^{-9} \text{ cm}^2) \cdot 20.83 = 4.89 \text{ MW}/\text{cm}^2$ a pulse peak intensity.

[R1] Hein, B., Willig, K. I., & Hell, S. W. (2008). Stimulated emission depletion (STED) nanoscopy of a fluorescent protein-labeled organelle inside a living cell. *Proceedings of the National Academy of Sciences*, 105(38), 14271-14276.

[R2] Willig, K. I., Harke, B., Medda, R., & Hell, S. W. (2007). STED microscopy with continuous wave beams. *Nature methods*, 4(11), 915-918.

Changes:

Results, Photostability and stimulated emission saturation intensity of the dye, Paragraph 3, (Page 6)

However, the squaraine-STED dye has a lower saturation intensity at 0.893 mW, which is ~ 3.4 -fold lower than that of ATTO 647N (3.069 mW, Figure 1c).

Changed to:

However, the squaraine-STED dye has a saturation intensity of **4.37 MW/cm²**, which is ~ 3.4 -fold lower than that of ATTO 647N (**15.0 MW/cm²**, **Figure 1c**, which is close to the data **10 MW/cm²** in reference²⁴).

Revised Figure 1:

[REDACTED]

Figure 3 Development of a highly photostable bright cell-permeable enhanced squaraine probe for mitochondria inner membrane labeling. (a) Chemical structure of MitoESq-635 used for the specific labeling of VDPs. (b) Absorption and emission spectrum of the enhanced squaraine dye, for which a 775 nm pulse laser can be employed for depletion and a 635 nm laser for excitation during STED setup. (c) The detected fluorescence signal of MitoESq-635 and ATTO 647N solution dye pool on coverslip as a function of the depletion beam intensity; the excitation beam was 635 nm with a 5 ps pulse width and 80 MHz, and the STED beam was 775 nm with 600 ps pulse width and 80 MHz. (d) Comparison of the photostability under a confocal laser microscope of HeLa cells co-stained with MitoESq-635 (0.1 μM) and MitoTracker Green (Rhodamine123) (0.1 μM). Upper row, MitoESq-635, excited at 633 nm, and collected at 645-680 nm. Lower row, MitoTracker Green (Rhodamine123), excited at 488 nm, and collected at 500-560 nm. Confocal images are under the same imaging conditions, excitation under 1.97 μW averaged power, 1 frame per second acquisition speed (imaging time: 0.66s, recovery time: 0.34s). Fluorescence signal of each image are plotted as a function of the recorded image number. Scale bar, 10 μm . (e) Comparison of the photostability under a STED nanoscope of living HeLa cells stained with MitoESq-635 (0.1 μM) and MitoTracker DeepRed (0.1 μM), respectively. STED images are under the same imaging conditions, excitation under 1.1 μW averaged power at 640 nm, STED beam of 30.2 mW average power at 775 nm, 1 frame per second acquisition speed (imaging time: 0.66s, recovery time: 0.34s). Fluorescence signal of each image are plotted as a function of the recorded image number. Scale bar, 2.5 μm .

Response to comment 8

Comment: *Can the authors please state how they assess viability (Suppl. Fig. 8, see my comment in the original referee report)? The data in Suppl. Fig. 8 is meaningless if the authors don't give the details on how they perform such an analysis.*

Response: Thank you for your suggestions. We have added the detailed explanation on the cell viability assessment and experimental procedures in the revised Supplementary Figure 9. The experimental details are listed as follows.

Added:

Detail for Supplementary Figure 9

In this experiment, a commercial chemical reagent of CCK-8, being nonradioactive, allows sensitive colorimetric assays for the determination of the number of viable cells in cell proliferation and cytotoxicity assays. Herein, we use the products from Dojindo (CCK-8) to carry out the cytotoxic assay. The detailed experimental procedures are listed as follows:

100 μ L of HeLa cell suspension (about 5000 cells/well) was firstly dispensed in a 96-well plate, and was pre-incubated for 24 hours in a humidified incubator (37°C, 95% humidity, 5% CO₂). Then 10 μ L of various concentrations of MitoESq-635 (final concentrations are 0, 0.5, 1, 2, 5 μ M) were added to the 96-well plate to be tested, which the 96-well plate was incubated for different time (1, 3, 6 hours) in the incubator. And then 10 μ L of CCK-8 solution was carefully added to each well of the above plate to avoid introducing bubbles into the wells. The obtained 96-well plate was incubated for another 3 hours in the incubator. And finally, the absorbance of the samples at 450 nm were measured using a microplate reader (Rayto RT-6100, Shenzhen, China).

$$\text{Cell Viability} = [(As-Ab) / (Ac-Ab)] \times 100\%$$

where As is absorbance of experimental wells (containing medium, CCK-8, MitoESq-635); Ac represents the absorbance of control wells (containing medium, CCK-8); Ab stands for the absorbance of blank wells (containing CCK-8).

Response to comment 9

Comment: *The new section on data processing is not understandable.*

Response: Thank you for your comments and sorry for our inaccurate statement. We revised it in the revised manuscript. You can find the revised section as follow.

Changes:

Methods and materials, Data processing, Page 16

“Data processing - In Figure 3, for box plot of width of mitochondria, each frame has 6 data points, which means 6 mitochondria in that frame are analyzed. Their widths are detected by using intensity profile in Fiji. The figure is plotted using Origin software, Plot-Statistics-Box Chart. For fluorescence intensity, using the same original images data with width mitochondria, choose 6 areas (10 pixels x 10 pixels) in the frames. In Fiji, Images – Stacks – Measure Stacks, then the total intensity of every square (10 pixels x 10 pixels) can be obtained in all frames. In every frame, we have 6 intensity data (6 squares), calculated the mean value intensity of every frame. To normalize the intensity, all intensities are divided by the mean value of the intensity of first frame. Then calculated the mean value and variance of intensity in each frame by using Statistics on Rows in Origin software. For resolution, intensity is obtained using Fiji, and Gaussian Cumulative Fit Peak is processed with Origin software.”

Changed to:

“Statistics analysis - Width of mitochondria, and fluorescence intensity are detected by using Fiji software. Both Box plot chart of width of mitochondria and error bar chart of fluorescence intensity are obtained by using Origin software. Gaussian Cumulative Fit Peak is processed with Origin 2018 software.”

Response to comment 10

Comment: *The caption in suppl. fig. 9 and 10 does not explain how cells were labelled. It would be good to have such information directly in each figure caption.*

Response: Thank you for your suggestions. We added how cells were labelled in Suppl. Fig. 12 (was Suppl. Fig. 10). We delete the original Suppl. Fig. 9, because we are not interested in the influence of excitation beam. In Suppl. Fig. 12, we use 0.5 μ M MitoESq-635 here, and the probe was subject to be incubated with HeLa cells for 30 minutes at 37 °C before imaging.

Supplementary Figure 12

Bursting of MitoESq-635 ($0.5 \mu\text{M}$) in HeLa cells caused by High dose depletion beam power in STED microscopy. The probe was subject to be incubated with HeLa cells for 30 minutes at 37°C before imaging. Depletion beam power, 51 mW. The molecules in red circle of STED image are bursting induced by high dose depletion beam, excitation wavelength: 633 nm, STED wavelength: 775 nm. Scale bar, $5 \mu\text{m}$.

Changes:

Caption of Supplementary Figure 13 (old Supplementary Figure 10)

Bursting caused by high dose depletion beam power in STED microscopy. Depletion beam power, 51 mW. The molecules in red circle of STED image are bursting induced by high dose depletion beam. Scale bar, $5 \mu\text{m}$.

Changed to:

Bursting of MitoESq-635 ($0.5 \mu\text{M}$) in HeLa cells caused by High dose depletion beam power in STED microscopy. The probe was subject to be incubated with HeLa cells for 30 minutes at 37°C before imaging. Depletion beam power, 51 mW. The molecules in red circle of STED image are bursting induced by high dose depletion beam, excitation wavelength: 633 nm, STED wavelength: 775 nm. Scale bar, $5 \mu\text{m}$.

Delete:

Original Supplementary Figure 9

Response to comment 11

Comment: *Suppl. Fig. 14: I cannot read the axes in panels d and e.*

Response: Thank you for your comments. They are “ $\tau_{\text{m}}=0-200$ [ps]” in panel d and “ $\tau_{\text{m}}=1000-3000$ [ps]” in panel e. We revised them in the revised text to make them clearer as shown below. (We reorder the

Supplementary Figure number according to when they occur in main manuscript. Now original Supplementary Figure 14 becomes new Supplementary Figure 7.)

Revised Suppl. Fig. 14:

Supplementary Figure 7

Optical properties of MitoESq-635 in different solvents and living cells

(a-b) Absorption and Fluorescence spectra of MitoESq-635 in different solvents (c) Fluorescence lifetime decay curves of MitoESq-635 in different solvents (d-g) FLIM imaging of Mito-ESq-635 in HeLa cell and PBS which was measured by DCS-120 time-correlated single photon counting equipment (DCS-120, Becker Hickl), excitation wavelength: 635 nm; detect range: 675/30 nm. On the mitochondria of Hela cells, the lifetime $T \approx 1.7$ ns. Meanwhile, the lifetime of MitoESq-635 in PBS ($1 \mu\text{M}$) is $T_1 \approx 65$ ps.

Response to comment 12

Comment: *Whenever a nonlinear lookup table is given, a colour bar for the lookup table should be provided. This also applies to the 2D histograms in suppl. figs. 1-3.*

Response: Thanks for your suggestions. We added colour bar in revised text. You can find it below as well.

Revised SI Figure 1-3:

Supplementary Figure 1

Co-localization experiment in U2OS cells employing MitoTracker Green as golden standard for mitochondrial marker; the first column, enhanced squaraine dye (20 nM) labeled U2OS cells; the second column, MitoTracker Green (50 nM) labeled U2OS cells; the third column, merged images; the fourth column, 2D intensity histogram and Pearson's coefficients; excitation wavelength: MitoESq-635 (640 nm), MitoTracker Green (490 nm); detect range: MitoESq-635 (660-740 nm), MitoTracker Green (495-575 nm); Scale bar, 5 μ m.

Supplementary Figure 2

Co-localization experiment in HeLa cells employing MitoTracker Green as golden standard for mitochondrial marker; the first column, enhanced squaraine dye (20 nM) labeled HeLa cells; the second column, MitoTracker Green (50 nM) labeled HeLa cells; the third column, merged images; the fourth column, 2D intensity histogram and Pearson's coefficients; excitation wavelength: MitoESq-635 (640 nm), MitoTracker Green (490 nm); detect range: MitoESq-635 (660-740 nm), MitoTracker Green (495-575 nm); Scale bar, 5 μ m.

Supplementary Figure 3

Co-localization experiment in SH-SY5Y and SKOV cells employing MitoTracker Green as golden standard for mitochondrial marker; the first column, enhanced squaraine dye (20 nM) labeled SH-SY5Y and SKOV cells; the second column, MitoTracker Green(50nM) labeled SH-SY5Y and SKOV cells; the third column, merged images; the fourth column, 2D intensity histogram and Pearson's coefficients; excitation wavelength: MitoESq-635 (640 nm), MitoTracker Green (488 nm); detect range: MitoESq-635 (700/75 nm), MitoTracker Green (525/50 nm); Scale bar, 10 µm.

Response to comment 13

Comment: *Suppl. Fig. 2: It appears that slightly different focus positions are imaged in the two colour channels, lowering overlap.*

Response: Thanks for your comments. Firstly, as you mentioned, the lower overlap can be caused by that a slightly different focus positions are imaged in the two colour channels, which could be chromatic aberration. Secondly, it seems that the Mitotracker Green aggregate at some other organelles as well, such as some small bubbles which are not mitochondria, but possibly lysosome.

Response to comment 14

Comment: *Hemozoin is a crystalline substance, such that "melting" is more appropriate there than for dye molecules.*

Response: Thank you for your explanation. We agree with your comment. We revised “melting” into “oxidized” in our revised manuscript.

Reviewer #3

Response to comment 13

Comment: *After revision, I recommend this manuscript to be published.*

Response: Thank you for your recognition on our revised manuscript.